# The chronODE framework for modelling multi-omic time series with ordinary differential equations and machine learning

Beatrice Borsari [1,2,6], Mor Frank [1,2,6], Eve S. Wattenberg [1,2], Ke Xu [3], Susanna X. Liu[1,2], Xuezhu Yu[1,2] & Mark Gerstein [1,2,3,4,5] ✉

Many genome-wide studies capture isolated moments in cell differentiation or organismal development. Conversely, longitudinal studies provide a more direct way to study these kinetic processes. Here, we present an approach for modeling gene-expression and chromatin kinetics from such studies: chronODE, an interpretable framework based on ordinary differential equations. chronODE incorporates two parameters that capture biophysical constraints governing the initial cooperativity and later saturation in gene expression. These parameters group genes into three major kinetic patterns: accelerators, switchers, and decelerators. Applying chronODE to bulk and single-cell time-series data from mouse brain development reveals that most genes (~87%) follow simple logistic kinetics. Among them, genes with rapid acceleration and high saturation values are rare, highlighting biochemical limitations that prevent cells from attaining both simultaneously. Early- and late-emerging cell types display distinct kinetic patterns, with essential genes ramping up faster. Extending chronODE to chromatin, we find that genes regulated by both enhancer and silencer *cis*-regulatory elements are enriched in brain-specific functions. Finally, we develop a bidirectional recurrent neural network to predict changes in gene expression from corresponding chromatin changes, successfully capturing the cumulative effect of multiple regulatory elements. Overall, our framework allows investigation of the kinetics of gene regulation in diverse biological systems.

During organismal development, changes in gene expression drive cell fate decisions, and activation or repression of genes at inappropriate times can disrupt normal cellular functions and lead to disease[1–4]. Epigenetic mechanisms play a critical role in regulating such changes[5]. Understanding the kinetics of gene regulation—the rate at which regulatory elements influence changes in gene expression over time—is the key to uncover mechanisms underlying both physiological processes and disease. Moreover, insights into regulatory kinetics can

provide therapeutic opportunities to control gene activation or silencing in precise, targeted ways[6–9].

Time-resolved genomic studies are a powerful way to study the kinetics of gene expression and its regulation at the epigenetic level. By capturing how gene expression and chromatin states change over time, it is possible to identify regulatory patterns that cannot be inferred from static snapshots. However, analyzing time-series genomic data poses challenges due to the inherent noise and

[1]Program in Computational Biology and Biomedical Informatics, Yale University, New Haven, CT, USA. [2]Department of Molecular Biophysics and Biochemistry, Yale University, New Haven, CT, USA. [3]Department of Computer Science, Yale University, New Haven, CT, USA. [4]Department of Statistics and Data Science, Yale University, New Haven, CT, USA. [5]Department of Biomedical Informatics and Data Science, Yale University, New Haven, CT, USA. [6]These authors contributed equally: Beatrice Borsari, Mor Frank. ✉e-mail: mark@gersteinlab.org

interconnections between signals, which can obscure the identification of meaningful biological insights. Modeling such data in an effective way requires approaches that not only reduce technical noise but also incorporate the biological principles underlying these molecular processes.

Epigenetic processes, such as chromatin accessibility and histone modifications, are shaped by two fundamental properties: cooperativity and saturation, both governed by fundamental physical and chemical constraints, related to the characteristics and quantities of nucleosomes, transcription factors (TFs), and other molecular components. Cooperativity refers to the self-reinforcing nature of molecular interactions. For instance, when chromatin starts to open, DNA tends to unwrap more easily from nucleosomes the more unwrapped it already is[10]. Similarly, once deposited at a specific locus, post-translational modifications of histones (e.g., methylation or acetylation) can rapidly propagate to neighboring histones[11–13]. On the other hand, these processes are characterized by natural limits, causing them to reach saturation over time. For example, chromatin accessibility or histone modifications are constrained by the number of nucleosomes present in a given region of the genome. Thus, once all nucleosomes in a locus have been modified or displaced, a saturation point (or carrying capacity) is reached. Notably, these properties also apply to gene expression kinetics. For example, the amount of RNA copies, transcriptional bursts, cooperative interactions among TFs, and feedback loop mechanisms contribute to changes in the kinetics of RNA production over time[14–18]. Understanding these shared principles can shed light on the temporal coordination between epigenetic mechanisms and gene expression.

Despite advances in deciphering gene expression kinetics—such as RNA production and degradation rates[19–22]—less is known about the kinetics of epigenetic processes and their impact on downstream gene expression over time. Chromatin accessibility signal is often used as a proxy for TF binding at regulatory elements, which, in turn, controls the expression of associated genes[23]. Bulk and single-cell (sc) multi-omics studies have linked chromatin accessibility to gene expression[24–26]. However, these studies are typically cross-sectional, capturing single time points rather than continuous temporal processes. Deciphering the kinetics of chromatin accessibility could

enable predictions of gene expression over time, but this requires computational methods capable of capturing various kinetic patterns of both epigenetic signals and genes.

To address these challenges, we developed chronODE, a multistep computational framework for integrating and modeling time-resolved multi-omics data (Supplementary Fig. 1). chronODE first generates smoothed time-series signals by fitting a logistic ordinary differential equation (ODE) to the raw genomic data. Using a single mathematical function, we can capture different kinetic patterns associated with both increasing and decreasing genomic signals (logistic, log-like, exponential, linear). Subsequently, chronODE integrates smoothed time-series signals from different modalities by learning their sequence-to-sequence temporal relationships through a bidirectional recurrent neural network (biRNN). We applied chronODE to time-series bulk and sc multi-omic data generated during mouse brain development[27–29]. We found that, during this process, gene expression programs are highly constrained, consistent with the principle of homeostasis governing neurogenesis[30]. Moreover, the kinetics of gene activation and repression are shaped by different factors, including cell-type specificity, gene essentiality, and regulatory mechanisms. Finally, we used our biRNN architecture to predict gene expression over time by modeling both enhancer- and silencer-mediated mechanisms of gene regulation. In doing so, we uncovered that proximal regulatory elements mainly drive the overall direction of gene expression changes (i.e., upregulation/downregulation), while distal elements play a stronger role in dictating their kinetic patterns.

## Results
### chronODE: modeling time-resolved genomic signals with the logistic function

Given the cooperative and saturating nature of genomic signals (Fig. 1), we propose modeling transcriptomic and epigenomic kinetics over time using the logistic function, previously applied in fields like bacterial growth[31,32]. In this context, the signal of a genomic locus (e.g., gene expression, chromatin accessibility, or histone modification) is defined as a time-dependent positive variable $z$, constrained within the interval $[a, b]$. The rate of change of $z$ over time $t$ can be studied using the following formula, which corresponds to the generalized form of

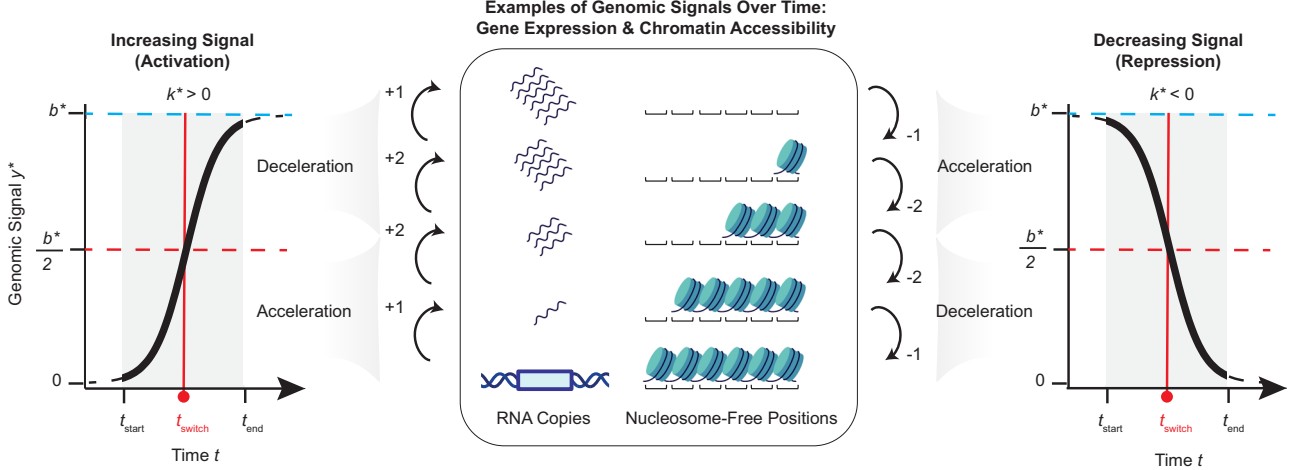

**Fig. 1 | Diagram illustrating how the logistic curve can be used to model time-series genomic signals. Left panel:** The curve represents a genomic signal ($y$ axis) undergoing a logistic increase over time ($x$ axis), modeled using the simplified form of the logistic ODE (Eq. 2) and with a positive growth rate constant (i.e., $k^* > 0$). Note that the genomic signal $y^*$ represents a translated and normalized version of the original genomic signal $z$. The curve is constrained between a lower asymptote of 0 and an upper asymptote defined by the parameter $b^*$ (dashed blue line). $t_{start}$ and $t_{end}$ represent the initial and final time points, respectively, monitored by the experimental time course. The curve is composed of an acceleration phase

followed by a deceleration phase, with the inflection point marking the transition between these phases at $t_{switch}$ (solid red line). This curve can be used to model changes in genomic signals over time, such as gene expression (e.g., mRNA copy number measured via RNA-seq) or chromatin accessibility (e.g., number of nucleosome-free positions measured via ATAC-seq or DNase-seq). The arrows indicate how the number of mRNA copies and nucleosome-free positions increases during the acceleration and deceleration phases. **Right panel:** Analogous representation for a signal undergoing a logistic decrease ($k^* < 0$). Elements of this figure were created in BioRender. Gerstein, M. (2025) https://BioRender.com/9ok1440.

the logistic ODE (see Supplementary Note 1, Propositions 1 and 5, and Supplementary Fig. 2):

$$\frac{dz}{dt} = k(z-a)\left(1 - \frac{z-a}{b-a}\right), \tag{1}$$

where $a$ and $b$ correspond to the lower and upper asymptotes, respectively, of the logistic curve in the real range of the data.

However, fitting this general form directly can lead to instability, particularly when the number of experimental time points available is limited[33]. To mitigate this and provide a simpler and easier-to-understand perspective, we fit the data using a simplified form of the ODE (after appropriate translation and normalization of the data; see Supplementary Note 1, Propositions 6–9, and Supplementary Fig. 2):

$$\frac{dy^*}{dt} = k^* y^* \left(1 - \frac{y^*}{b^*}\right), \text{ with } y^*(t_{\text{start}}) = y^*_{\text{start}} \tag{2}$$

where $t_{\text{start}}$ represents the initial time point of the experimental time course, and $y^*_{\text{start}}$ represents the normalized gene expression level or chromatin signal of the genomic locus at $t_{\text{start}}$.

We first solve Eq. 2 analytically to obtain an explicit formulation for $y^*$ (see Supplementary Note 1, Proposition 1, for the derivation of the analytical formula):

$$y^*(t) = \frac{b^* C^* e^{k^* t}}{b^* + C^* e^{k^* t}}, \quad \text{where}$$
$$C^* = \frac{y^*_{\text{start}}}{\left(1 - \frac{y^*_{\text{start}}}{b^*}\right) e^{k^* t_{\text{start}}}}. \tag{3}$$

Then, we numerically estimate the two kinetic parameters $k^*$ and $b^*$ that best fit the data. Specifically, our proposed chronODE package optimizes this solution by repurposing a generic initial value problem solver and constraining the numerical solution to the logistic space, so that parameters $k^*$ and $b^*$ (and their corresponding $k$ and $b$ values back-transformed to the original range) can be easily interpreted. In this context, $b^*$ is a positive number and corresponds to the saturation level of the normalized logistic curve, i.e., the horizontal asymptote that approximates the maximum predicted level of $y^*$, while $k^*$ represents the growth/decay rate constant, which dictates how fast the signal ramps up or slows down over time. In the case of increasing (i.e., activated) profiles, $k^* > 0$, whereas decreasing (i.e., repressed) profiles have $k^* < 0$ (Fig. 1).

We describe the logistic curve as consisting of two biologically meaningful phases. First, the acceleration phase (where $|y^{*\prime}|$ progressively increases over time) models the cooperative aspect of epigenetic and transcriptional processes, which tend to occur more frequently and easily after the initial state[10–14,16]. Second, the deceleration phase (where $|y^{*\prime}|$ progressively decreases) models the saturating aspect of these processes (Fig. 1). We define $t_{\text{switch}}$ as the time point that marks the transition between these two phases as follows:

$$t_{\text{switch}} = \frac{1}{k^*} ln\left(\frac{b^*}{y^*_{\text{start}}} - 1\right) + t_{\text{start}}. \tag{4}$$

Equation (1) not only models the kinetics of genomic signals while accounting for the underlying biophysical and biochemical constraints, but it also provides the flexibility to capture diverse patterns beyond a standard logistic curve. In particular, it allows for variability in the behavior of genes or regulatory elements, accommodating patterns such as exponential, linear, or logarithmic-like trajectories, which correspond to the accelerating, switching (inflection point), and decelerating phases of the logistic curve, respectively. In essence, it does not assume that all signals strictly follow a full logistic profile.

We note that this approach is well-suited for modeling genomic signals that exhibit monotonic increases or decreases. While these monotonic fits describe most of the observed transcriptional programs, a small fraction of genes may display peak-like expression profiles over time, such as those involved in circadian clocks, cell cycle regulation, or stress responses[34–36]. Thus, we further extended our methodology by introducing a piecewise fitting approach that can accommodate such profiles by combining two logistic curves (or two portions of them). Although alternative functions (e.g., Gaussian curves) could be employed for these cases, our choice to use logistic functions maintains consistency within the framework and allows for comparable kinetic parameters across both monotonic and piecewise fits. Both approaches are implemented as standalone pipelines, providing a scalable solution for genome-wide applications at both bulk and sc levels.

## Kinetic profiling of the developing mouse brain transcriptome reveals a balance between rate and saturation level

We used chronODE to investigate the kinetics of gene expression regulation during mouse brain development. We first analyzed time-series RNA-seq maps generated for three regions of the mouse brain (forebrain, midbrain, and hindbrain) across eight developmental time points, from post-conception (PC) day 10.5 to the first postnatal day (PN) (Supplementary Table 1)[27]. On average across the three regions, we identified >12,000 genes differentially expressed (DE) over time (Supplementary Fig. 3A). We applied chronODE on the set of DE genes (defining, for a given gene, $t_{\text{start}} = 10.5$ and $y^*_{\text{start}} =$ normalized expression level of the gene at time point 10.5; see "Methods"), and identified an average of 87% monotonic fits, suggesting that the logistic ODE is appropriate for modeling the majority of time-series patterns in our dataset (Supplementary Fig. 3B). Conversely, a small fraction of genes (average of 8%) displaying peak-like patterns were captured using piecewise fits (Supplementary Fig. 3C–F). The remaining ~5% of genes exhibited complex or variable patterns that were not well-suited for fitting with the current models.

When analyzing the distribution of $k$ and $b$ values across genes and brain regions, we found that the monotonic genes are clustered into three major quadrants based on their $k$ & $b$ combinations: high $k$ & low $b$ (Q1), low $k$ & low $b$ (Q2), and low $k$ & high $b$ (Q3) (Fig. 2A). This distribution exhibits an L-shaped pattern (Pearson's $r$ correlation = −0.44, $p$ value < 2.2e-16), where high values of both $k$ and $b$ are not observed together (i.e., Q4 is absent). A closer examination revealed that Q1 genes (27%) change expression faster, with expression profiles following the full logistic curve (Fig. 2B). In contrast, Q2 genes (59%) and Q3 genes (14%) align with only portions of the logistic curve and are characterized by slower kinetics. Specifically, Q2 genes approach saturation, while Q3 genes remain far from saturation throughout the observed time frame.

Within these slowly saturating Q3 genes, those repressed during brain development are associated with nucleotide biosynthesis and amino acid metabolism—metabolic processes essential for early cell proliferation but downregulated later on during lineage commitment[37,38] (Supplementary Fig. 4A). This result is consistent with these genes being already far from saturation by day 10.5 (Fig. 2B). In contrast, Q3 activated genes are enriched for functions related to oxidative phosphorylation and mitochondrial ATP synthesis (Supplementary Fig. 4B). These key components of aerobic respiration become increasingly prominent at later developmental stages, particularly during the peri- and post-natal periods[39], aligning with the slower activation and delayed saturation of Q3 activated genes (Fig. 2B).

We performed a similar analysis for genes with peak-like expression profiles, and found that, also in this case, only a small fraction (~8%) corresponds to Q4 genes (Supplementary Fig. 4C–F). Altogether, this suggests that genes cannot undergo rapid expression changes while simultaneously achieving high saturation. Consequently, the

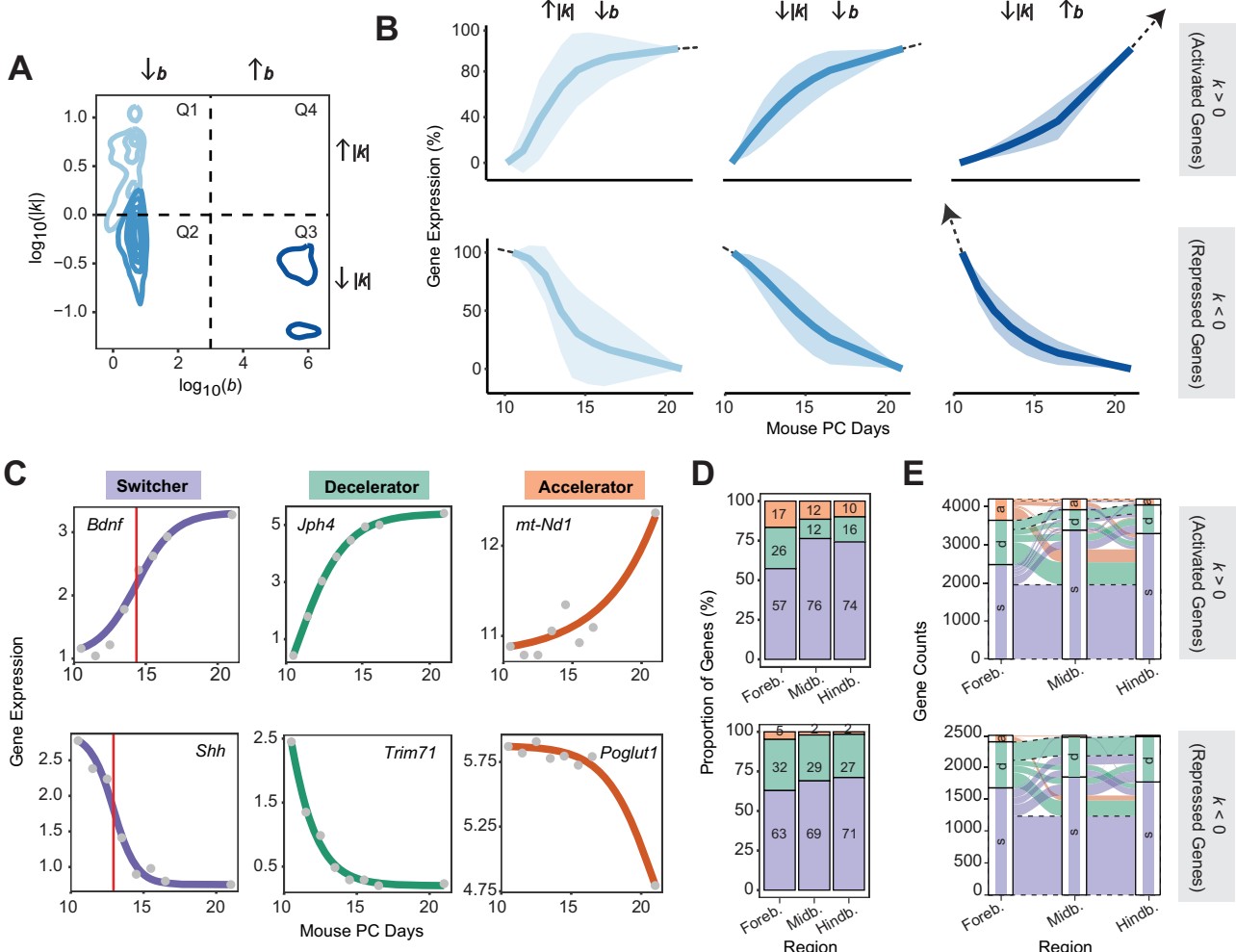

**Fig. 2 | Kinetic characterization of the developing mouse brain transcriptome across three brain regions. A** Distribution of the kinetic parameters $k$ (magnitude expressed in absolute value, $y$ axis) and $b$ ($x$ axis) across all monotonic genes in the three brain regions. Genes are grouped into three quadrants (Q1–Q3) based on k-means clustering of their $k$ & $b$ combinations. No genes are found in Q4. **B** Lineplot showing, for each set of activated and repressed genes in Q1 through Q3 groups, the average expression ($y$ axis) over time ($x$ axis; PC post-conception). Note that the mean expression levels (expressed in $\log_2$(TPMs)) were rescaled to the range 0–100% to allow for comparison across Q1–Q3 groups. The error band corresponds to the standard deviation of the mean rescaled expression level (i.e., mean ± SD). Q1 activated and repressed genes have an average expression profile that resembles the full logistic curve, whereas Q2 and Q3 genes align with the decelerator and accelerator parts of the curve, respectively. The arrow in Q3 genes

indicates that these profiles remain far from reaching the saturation level $b$. **C** Examples of brain-related genes whose expression profile follows switcher, decelerator, and accelerator profiles, either increasing (upper panels) or decreasing (lower panels). Colored lines correspond to chronODE's fitted curve reconstructed using the kinetic parameters $a$, $b$, and $k$ (see also Supplementary Fig. 2); gray dots correspond to raw data points. The red vertical line indicates the switching point ($t_{switch}$). **D** For each brain region ($x$ axis), proportion (%) of genes ($y$ axis) characterized by switcher (purple), decelerator (green), and accelerator (orange) profiles. **E** Number of genes ($y$ axis) that show concordant (dashed black line) and discordant (continuous colored line) kinetic patterns across the three regions ($x$ axis) (s switcher, d decelerator, a accelerator). In this case, we focused on the set of 4204 and 2489 genes that have consistently increasing (upper panel) or decreasing (lower panel) monotonic fits in all three regions, respectively.

expression level of rapidly saturating genes is fundamentally constrained (Q1 genes).

Although most genes follow monotonic expression patterns, our results indicate that not all of them transition from full inactivation to full activation (or vice versa) at the same pace (Fig. 2B), and that these differences may correlate with the timing of when their functions are required. To systematically capture the temporal progression of gene expression kinetics, we developed a classification scheme that directly ties the gene's kinetic profile, as dictated by the $k$ and $b$ parameters, to the specific time window defined by $t_{start}$ and $t_{end}$ (in our case, days 10.5 and PN, respectively). This classification identifies the switching time ($t_{switch}$) of the gene—the point at which its expression shifts from accelerating to decelerating (see Eq. 4)—and determines whether $t_{switch}$ occurs before (decelerator), within (switcher), or after (accelerator) the time window monitored by the time-series study (Fig. 2C,

Supplementary Fig. 5A, and Supplementary Data 1). This allows us to describe the full signal curve in the context of the time window [$t_{start}$, $t_{end}$] (Supplementary Fig. 5B, C). Switcher profiles were the most abundant among both activated and repressed genes, while only a small fraction of genes were classified as accelerators (Fig. 2D). Moreover, although gene activation and repression were largely conserved, with nearly 100% of the genes showing either always an increasing or always a decreasing trend across the three brain regions, their kinetic patterns varied significantly. In fact, 47% and 40% of activated and repressed genes, respectively, were assigned to a different kinetic category in at least two regions (Fig. 2E).

Overall, chronODE allowed us to decipher the kinetics of gene expression during brain development at the resolution of individual genes, highlighting how changes in gene expression result from a balance between the rate of expression and the extent of saturation.

## Cell type emergence and gene essentiality shape gene expression kinetics

After mathematically quantifying the kinetics of gene expression during brain development, we next sought to uncover their biological meaning. Our classification indicates that the kinetic patterns of certain genes differ substantially between the forebrain, midbrain, and hindbrain (Fig. 2E). We hypothesized that these variations could be influenced by the cellular composition of each brain region and the changes in this composition over time. In fact, each region is composed of specific cell types, with specific characteristics, that emerge at different developmental stages, potentially shaping the kinetics of gene expression[40]. To test this hypothesis, we extended our analysis from tissue-level data to sc resolution. Specifically, we applied our chronODE framework to a recently published time-series scRNA-seq atlas of the developing mouse embryo[29]. This dataset includes 41 brain-related cell types that emerge at various time points between embryonic days E8.5 and E14.25. For our subsequent analyses we focused specifically on activated genes, since they were detected across all cell types, in contrast to repressed genes which were mostly restricted to cell types appearing between E8.5 and E9 (Supplementary Fig. 6A).

Genes with distinct kinetic behaviors, as defined by the three quadrants in Fig. 2A, were observed also across most individual cell types, consistent with the diversity of gene expression kinetics identified in the bulk tissue analysis (Supplementary Fig. 6B). However, the L-shaped distribution of these three kinetic subpopulations changed markedly over time as new cell types emerged. Stratifying cell types into five temporal groups revealed that genes become progressively constrained to a smaller range of $k$-$b$ combinations at later developmental time points (Fig. 3A). In other words, late cell types, such as amacrine precursors and cholinergic amacrine cells, show more constrained kinetic patterns of gene activation, compared to early cell types (e.g., neural progenitors and motor neurons) (Supplementary Fig. 6B). Moreover, when taking a closer look at their temporal progression, we found that genes activated in early-appearing cell types predominantly followed decelerating profiles and were already close to their saturation level $b$ by day E8.5 (Fig. 3B). In contrast, genes activated in late-emerging cell types displayed mostly switching profiles and, to a lesser extent, accelerating profiles. These findings confirmed our initial hypothesis, suggesting that the timing of cell type appearance seems to play a critical role in shaping gene expression kinetics.

Still, among genes activated in the early cell types, we observed a fraction following slower, accelerating profiles (average of 11% in E8-E9 cell types; Fig. 3B). This suggests that even though these cells are present from early stages of development, they selectively express certain genes at slower rates. This observation prompted us to investigate additional factors that might contribute to the regulation of gene expression kinetics. In particular, we hypothesized that a cell may tend to prioritize the expression of genes that are essential for its survival. To this end, we integrated in our sc analyses a recently published catalog of essential genes[41]. We observed that, especially in the earliest cell types (E8-E9 appearance days), essential genes tend to ramp up faster (higher $k$ values) and are closer to their saturation levels (i.e., higher proportion of decelerator and switcher profiles) compared to non-essential genes (Fig. 3C, D). Thus, our kinetic analysis provides an alternative perspective: the essentiality of a gene for cell viability likely influences its expression kinetics, determining how rapidly it is expressed.

## Genes linked to poly-pattern cCREs are more dynamic during brain development and are enriched in brain-specific functions

Besides estimating gene expression kinetics, chronODE is also suitable for studying the kinetics of chromatin activity at regulatory elements in the genome, providing insights into gene regulatory mechanisms. To this end, we also analyzed the kinetics of chromatin accessibility at candidate *cis*-regulatory elements (cCREs) from the ENCODE project[42].

Specifically, we applied chronODE to time-series DNase-seq and ATAC-seq maps[28] available for the same time points and brain regions as the bulk RNA-seq data (Supplementary Tables 2 and 3). We identified cCREs with both increasing ($k > 0$) and decreasing ($k < 0$) trends, and classified their kinetics in each of the three regions (Supplementary Data 1). We then investigated how activated and repressed cCREs relate to the expression of target genes over time. To achieve this, we employed a simple proximity criterion to pair mouse cCREs with target genes by linking each cCRE to its nearest gene based on linear distance[43,44]. Although this is a simplified approach, we note that our framework could also accommodate more complex linking methods, such as those recently applied by ENCODE in the context of the human genome[45]. Our analysis revealed that genes are not only linked to multiple cCREs, as previously reported[44], but that these cCREs also exhibit distinct regulatory roles. Specifically, on average 50% of genes are regulated by both activated ($k > 0$) and repressed ($k < 0$) cCREs (Supplementary Fig. 7A). We refer to these genes, which display mixed cCRE trends, as poly-pattern (poly-p.) genes (Fig. 4A). In contrast, mono-pattern (mono-p.) genes are exclusively associated with a single type of cCRE trend—either all activated or all repressed.

Across all three brain regions, genes with poly-pattern regulation were linked to more distal cCREs and exhibited larger expression changes over time compared to mono-pattern genes (Fig. 4B, C). This suggests that poly-pattern genes play a more significant role in brain development, as they are enriched in neural and brain-specific functions, including neurogenesis, trans-synaptic signaling, and sensory system development (Fig. 4D). In contrast, mono-pattern genes were predominantly associated with housekeeping functions such as gene expression and compound biosynthesis. A central question is what drives the highly dynamic behavior and specialized functions of poly-pattern genes: is it the number of regulatory elements or the diversity of their regulatory trends? To address this, we disentangled the contributions of cCRE count and trend diversity (mono- vs. poly-pattern regulation). Our analysis demonstrated that both the number of cCREs and the diversity of their trends significantly correlate with a gene's dynamic expression profile and its involvement in brain-specific functions (Supplementary Fig. 7B, C). This finding expands on our conclusion that gene expression is fundamentally a constrained process by having a balanced regulation involving both activation and repression mechanisms.

Overall, these results suggest that the precise expression of genes important for brain development may be governed by a more complex network involving regulatory elements with mixed trends. In contrast, those genes associated with non-brain-specific functions may rely on a simpler regulome, potentially reflecting a differential degree of control based on the biological significance and impact of these genes.

## Predicting time-series gene expression from chromatin signals of associated cCREs

The insights gained from our previous analyses improve our understanding of how gene expression evolves over time, particularly in terms of expression kinetics and the impact of epigenetic changes at regulatory elements. This knowledge can be applied to develop temporal models that predict gene expression at future time points, accounting for the additive effect of multiple cCREs associated with a given gene[44]. Beyond prediction, incorporating chromatin regulation into these models can also provide insights into the complex interplay between transcriptome and epigenome.

Towards this goal, the first step in our analysis was to examine the correlation, over time, between gene expression and chromatin accessibility at associated cCREs. We observed a bimodal distribution, with 66% of the gene-cCRE pairs showing positive correlation and 34% showing negative correlation (Supplementary Fig. 8A). In this context, we make the simple assumption that when a cCRE is open, TFs can access the gene, facilitating expression. In contrast, a closed cCRE

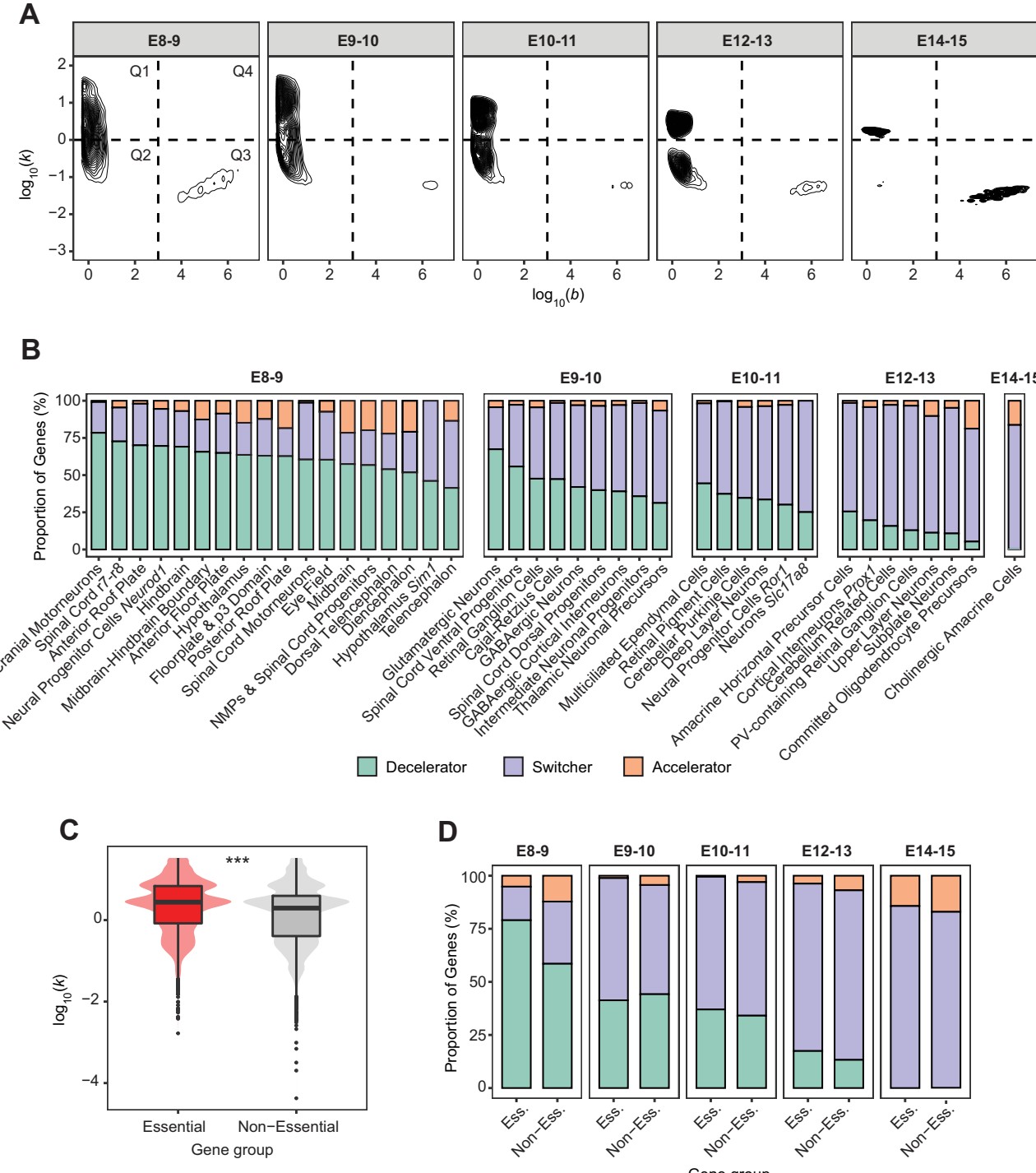

**Fig. 3 | Cell type specificity of gene expression kinetics. A** Distribution of the kinetic parameters $k$ ($y$ axis) and $b$ ($x$ axis) for activated genes ($k > 0$) across 41 brain cell types. Cell types have been grouped based on their day of appearance (between 8 and 9, 9 and 10, 10 and 11, 12 and 13, and 14 and 15 embryonic [E] days). **B** For each cell type ($x$ axis) within these five groups, proportion of genes (%; $y$ axis) that are characterized by switcher (purple), decelerator (green), and accelerator (orange) profiles. **C** Distribution of the kinetic parameter $k$ ($y$ axis) for essential (red) and non-essential (gray) genes ($x$ axis). Box plots present the median as the center, 25% and 75% percentiles as box limits, and whiskers extending to the largest and smallest values within the 1.5 × interquartile range (IQR) of the box limits. We applied the Wilcoxon Rank-Sum test (two-sided, unpaired) to compute significant differences (***: $p$ value < 0.001 [$p$ value = 4.2E-203]; $n$ = 54,212; we note these are unreplicated single-cell data from Qiu et al.[29]). **D** Proportion (%; $y$ axis) of essential (Ess.) and non-essential (Non-Ess.) genes ($x$ axis) belonging to the three kinetic classes (color-coded as in **B**).

restricts TF access and potentially reduces gene activity. Based on this assumption, the positively correlated cCRE-gene pairs were interpreted as cases where the cCRE acts as an enhancer of gene expression, since the accessibility of the cCRE aligns with the direction of gene expression. In contrast, the negatively correlated cCRE-gene pairs were interpreted as cases where the cCRE exerts a silencer effect on gene expression. This information is crucial for determining the direction of gene expression changes (activation or repression).

On the other hand, we found that the kinetics of gene expression were correlated with the diversity of cCRE regulation associated with

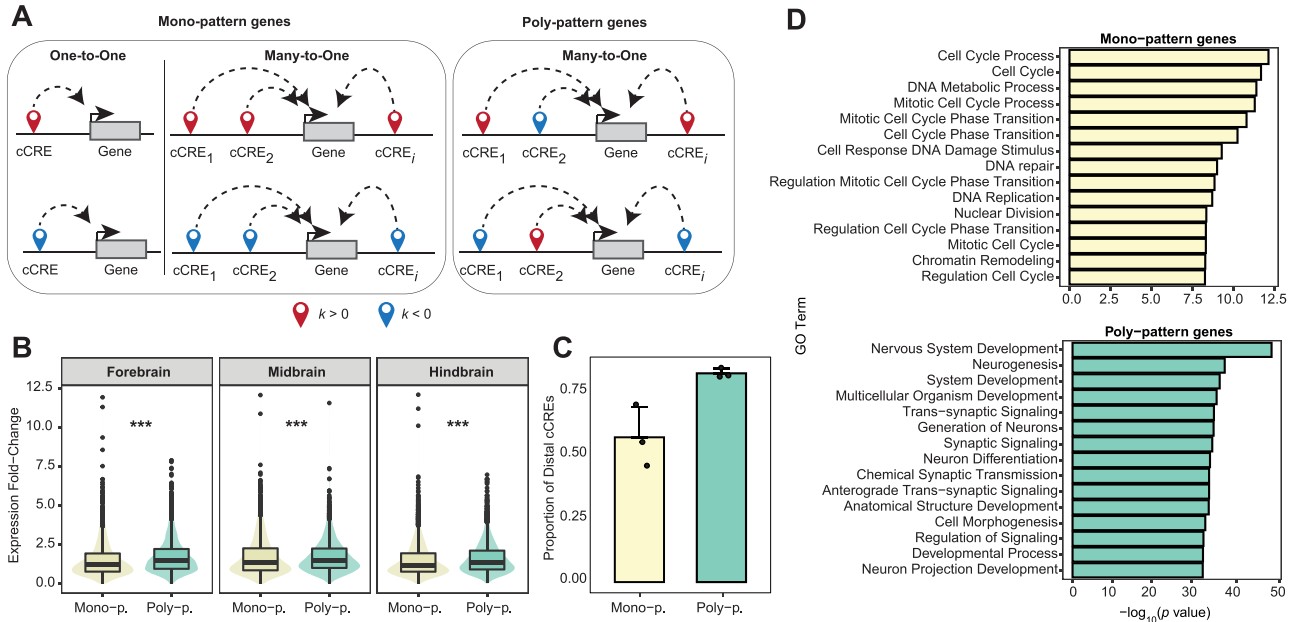

**Fig. 4 | Genes linked to poly-pattern cCREs are more dynamic during brain development and are enriched in brain-specific functions. A** Schematic illustrating the distinction between mono-pattern and poly-pattern genes. Mono-pattern genes (left panel) can be linked to one or more cCREs, but if associated with multiple cCREs, all of these cCREs exhibit the same regulatory profile—either all activated ($k > 0$, red, upper panel) or all repressed ($k < 0$, blue, lower panel). Genes linked to a single cCRE represent the simplest case of mono-pattern regulation. In contrast, poly-pattern genes (right panel) are associated with multiple cCREs, which may be predominantly activated (upper panel) or repressed (lower panel). **B** Distributions of gene expression fold-change ($\log_2$(TPM); $y$ axis) for mono-pattern (mono-p.) and poly-pattern (poly-p.) genes ($x$ axis) across the three brain regions. Box plots present the median as the center, 25% and 75% percentiles as box limits, and whiskers extending to the largest and smallest values within the $1.5 \times$ IQR of the box limits. We applied the Wilcoxon Rank-Sum test (two-sided, unpaired) to compute significant differences (***: $p$ value $< 0.001$). Forebrain: $p$ value $= 7.6E$-33 ($n = 7992$; 2 biological replicates); Midbrain: $p$ value $= 1.5E$-5 ($n = 5999$; 2 biological replicates); Hindbrain: $p$ value $= 9.5E$-16 ($n = 6772$; 2 biological replicates). **C** Proportion of distal cCREs ($y$ axis) linked to mono- and poly-pattern genes ($x$ axis). Distal cCREs are defined as those located >2 Kb from the nearest annotated transcription start site of the gene. We report mean proportion and standard deviation across the three brain regions ($n = 3$). Overlayed black dots indicate the proportion of distal cCREs in each of the three regions. **D** Gene Ontology biological process terms ($y$ axis) enriched among mono- and poly-pattern genes. Significance of GO terms' enrichment was determined with a hypergeometric test (two-sided). Only GO terms reporting a Benjamini-Hochberg adjusted $p$ value < 0.01 were considered. Among these, we show in the panel the 15 most significant GO terms with the corresponding $-\log_{10} p$ value ($x$ axis).

the gene (mono-p. *vs.* poly-p.; Supplementary Fig. 8B). Specifically, activated genes with decelerator profiles were more likely to undergo poly-pattern regulation compared to switchers and accelerators, while the opposite was observed for repressed genes. This suggests that the type of epigenetic regulation a gene undergoes plays a key role in shaping the kinetics of its expression profile, making it important to incorporate this information into the predictive model.

Building on these findings and utilizing our many-to-one linking schema—where multiple cCREs are linked to a given gene over time—we designed a model architecture (Fig. 5A) capturing four types of gene regulatory mechanisms that differ in the direction (enhancer/silencer) and diversity (mono-p./poly-p.) of cCRE regulation as follows: (1) mono-pattern genes subject to enhancer regulation (enhancer mono-p.), where all cCRE signals linked to a gene are positively correlated with its expression; (2) mono-pattern genes subject to silencer regulation (silencer mono-p.), where all cCRE signals linked to the gene are negatively correlated with its expression; (3) poly-pattern genes subject to enhancer regulation, where the majority of cCRE signals are positively correlated with gene expression, while a minor fraction exhibit silencer-like effects (enhancer poly-p.); and (4) poly-pattern genes subject to silencer regulation, where most cCRE signals are negatively correlated with gene expression, with a smaller proportion displaying enhancer-like effects (silencer poly-p.). This approach enabled us to train our model towards predicting both the direction and kinetic patterns of gene expression over time. Each subset of genes was used to train the model independently, with an 80:20 split for training and testing, respectively.

Our modeling approach predicts gene expression at each time point through regression, using a bidirectional RNN-based architecture, which is well-suited for time-series data. Given a multicCRE-gene pair $i$, we define the complete cCRE time-series input signal associated with a given gene as the matrix $\mathbf{C}_i$, such that:

$$\mathbf{C}_i = [\mathbf{c}_{i,0}, \mathbf{c}_{i,1}, \ldots, \mathbf{c}_{i,t}]. \tag{5}$$

At each time point $t$ ($t \in \{1, 2, \ldots, 8\}$) we define $\mathbf{c}_{i,t}$ as the vector of $m$ cCRE signals:

$$\mathbf{c}_{i,t} = [c_{i,t,1}, c_{i,t,2}, \ldots, c_{i,t,m}]. \tag{6}$$

$\mathbf{C}_i$ is the input used to predict $\mathbf{g}_i$, i.e., the gene expression profile over time:

$$\mathbf{g}_i = [g_{i,0}, g_{i,1}, \ldots, g_{i,t}]. \tag{7}$$

We use a biRNN to model gene expression $g_{i,t}$ at a particular time point $t$, as a function of the cCRE chromatin accessibility signal vector $\mathbf{c}_{i,t}$. In fact, capturing the interplay between chromatin and gene expression is not a trivial task, since multiple scenarios are possible. For instance, changes in chromatin accessibility at cCREs upstream of the promoter can potentially influence gene expression, for example, by enabling TF binding or facilitating Pol II release. Conversely, transcriptional activity—through Pol II elongation—can also induce changes in chromatin accessibility at both upstream and downstream cCREs. The biRNN

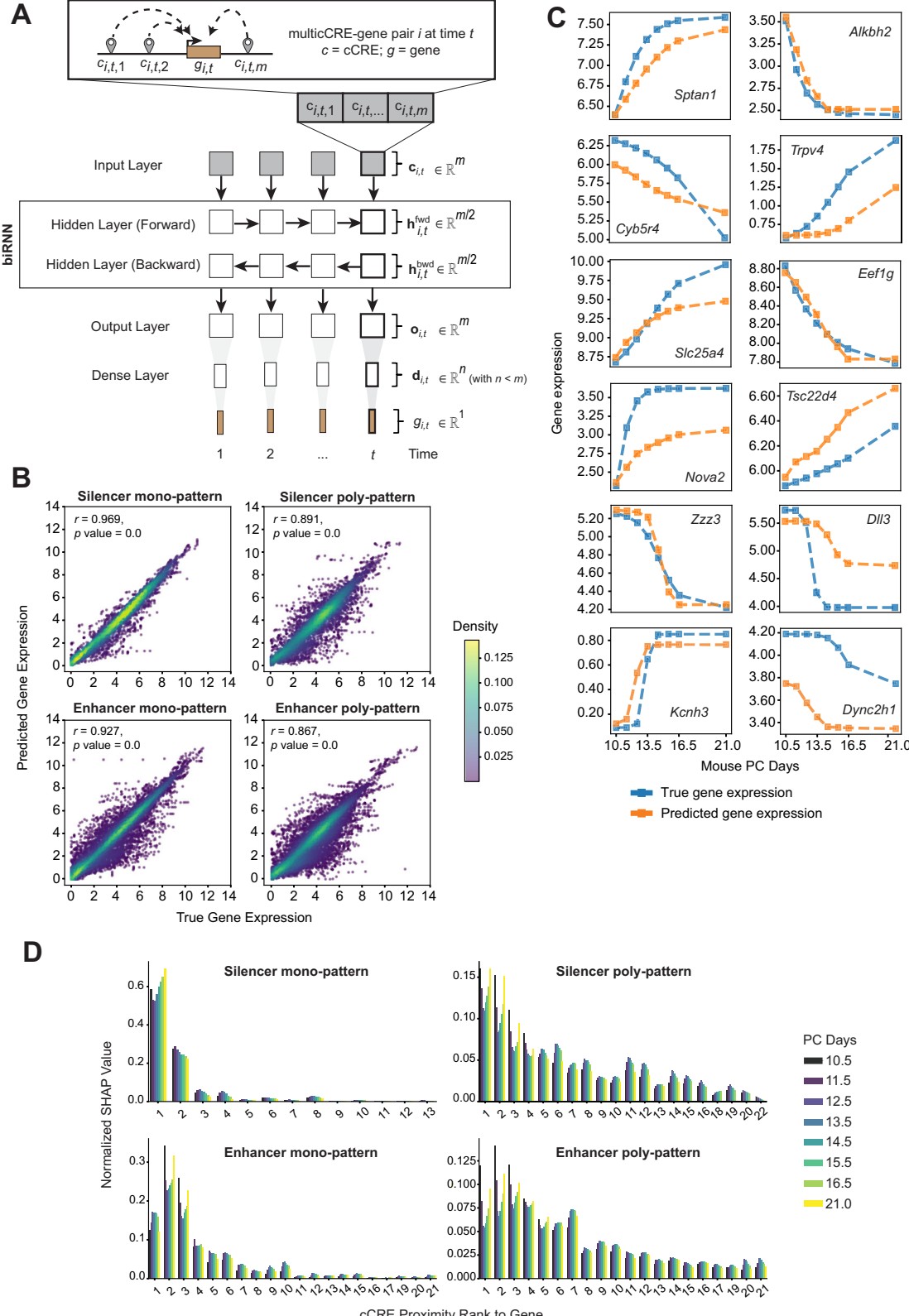

allows us to cover this wide range of scenarios by leveraging chromatin signals at both past and future time points to make predictions of gene expression. Specifically, we define the following relation:

$$g_{i,t} = \mathcal{D}(\,\text{biRNN}\,(\mathbf{c}_{i,t})), \tag{8}$$

where $\mathcal{D}$ represents a dense layer that takes as input the output of the biRNN.

This modeling approach effectively handles the highly dimensional nature of the omics data and enables accurate regression of gene expression levels exhibiting diverse kinetic patterns over time.

**Fig. 5 | Predicting time-series gene expression from chromatin signals of associated cCREs. A** Architecture of the biRNN-based model. At each time point, the input for a specific multicCRE-gene pair $i$ is a vector **c** of dimension $1 \times m$, where $m$ represents the number of cCRE signals associated with the gene. The model generates an output vector, representing the gene expression levels **g** across all time points. **B** Density plot showing cross-gene expression correlation for each of the four regulatory mechanisms. Each scatterplot shows true ($x$ axis) versus predicted ($y$ axis) expression values ($\log_2$-transformed TPMs) across all genes in the test set, alongside their degree of correlation as measured by Pearson's correlation coefficient ($r$) and the corresponding Benjamini–Hochberg adjusted $p$ value. **C** Representative examples of gene expression predictions over time

demonstrating our model's capability of capturing the three kinetic patterns (switchers, decelerators, and accelerators). True and predicted expression values are color-coded. Expression values correspond to $\log_2$-transformed TPMs. **D** Barplot showing, for each of the four regulatory mechanisms, the relative feature importance (computed using the SHAP method; $y$ axis), for different cCREs associated with genes in the test set ($x$ axis). The $x$ axis represents the proximity rank of each cCRE to its associated gene, with rank 1 indicating the closest cCRE. SHAP importance values are shown for each of the eight time points (color-coded). The analysis highlights that cCREs closer to the gene have higher predictive importance at the initial and final time points, while more distant cCREs show greater importance at intermediate time points.

We note that although here we focused exclusively on chromatin accessibility as input data, our modeling architecture is versatile and can potentially incorporate additional cCRE chromatin features (such as histone post-translational modifications) at each time point $t$ in the form of a tensor.

To evaluate the performance of the model, we first computed cross-gene correlations between true and predicted expression values. Our predictions on the test set showed a strong positive correlation ($r > 0.85$) in all four types of regulatory mechanisms (Fig. 5B), higher compared to previous reports of chromatin-gene expression predictions[46]. We also evaluated the model performance by computing the distribution of mean squared error (MSE) and cross-timepoint correlation between true and predicted gene expression values (Supplementary Fig. 8C). In particular, we achieved mean cross-timepoint correlations ranging between 0.57 and 0.90, higher than equivalent cross-cell type correlations reported previously[46]. To further showcase the predictive capability of our model, we also included some examples of true $vs.$ predicted gene expression time-series profiles (Fig. 5C). We note that the usage of the biRNN is especially effective for predicting the expression of poly-pattern genes, which are particularly challenging to model due to the mixed contributions of enhancer and silencer cCRE signals (Supplementary Fig. 8D). The biRNN leverages the more comprehensive learning context provided by both past and future time points to address this complexity. Additionally, incorporating a dense layer enhances the model's capacity to learn complex kinetic relationships between cCREs and their associated genes.

Finally, we investigated how the distance between cCREs and their target genes influences gene expression predictions using the SHapley Additive exPlanations (SHAP) method[47]. Specifically, we computed the average importance of each cCRE across all the samples in the test set. Our analysis revealed that closer cCREs have a greater impact on predicting gene expression, underscoring their biological importance in gene regulation, consistent with previous findings[43]. Notably, these proximal cCREs play a dominant role in predictions at the first and last time points, whereas middle-to-distant cCREs contribute more to the predictions at intermediate time points, thus directly influencing the kinetic patterns of the gene expression signals (Fig. 5D). This finding indicates that multiple cCREs provide the flexibility needed to capture complex kinetic patterns of gene expression. Altogether, this suggests that, during brain development, gene expression kinetics are shaped by the cumulative effects of numerous cCREs, especially the more distal ones.

## Discussion

Time-series functional genomic assays are a powerful way to investigate the kinetics of activation and repression of multiple genes and regulatory elements. However, characterizing these kinetics and linking them to biological processes over time poses significant challenges. Both epigenomic and transcriptomic datasets are inherently noisy and characterized by diverse kinetic patterns, making it difficult to discern major trends by simply comparing raw data. We showed that by using one logistic ODE we can model multiple kinetic patterns at the genome-wide scale, and we find that these kinetics are associated with

different cell types emerging over time. Specifically, this ODE effectively captures the cooperative and saturation-like behaviors of genomic signals, such as nucleosome remodeling, post-translational modification of histones, and gene expression. To facilitate the application of this ODE at the genome-wide scale, we developed chronODE, a software that integrates the ODE-fitting workflow into a single command, similar to other genomic tools that package established models into user-friendly software[48,49]. The fitting process uses a simplified form of the ODE with two parameters, instead of the generalized form of the ODE with three parameters, improving both interpretability and stability when working with a small number of data points. The chronODE framework can be used to quantitatively estimate where a gene is in its expression life cycle during a given process—whether it is close to saturation or in the acceleration phase of its expression, a characteristic that cannot be precisely quantified from the raw data and requires a rigorous modeling approach. To overcome the limitations of fitting ODEs with more than two parameters with a limited number of data points available, incorporating regularization may offer a promising direction to enhance both flexibility and stability.

This methodology offers several advantages. First, it uses a simple and intuitive mathematical function to model time-series genomic signals. Prior approaches to model this type of data have often relied on higher-order polynomial functions (e.g., B-splines), which are difficult to interpret because each gene is fitted with a unique set of base functions optimized to match data trends, lacking a direct connection with the underlying biology[50,51]. In contrast, the logistic ODE yields two intuitive and interpretable kinetic parameters with direct biological meaning.

To demonstrate its utility, we applied chronODE to bulk and sc multi-omic datasets from mouse brain development. The ability to fit the expression profiles of most genes with a logistic function suggests that gene expression is a smooth process characterized by gradual changes over time. Moreover, the observation that logistic parameters exhibit an L-shaped distribution emphasizes that gene expression is fundamentally constrained, similar to other chemical and biological phenomena[52,53]. In particular, we found that it is almost impossible for genes to quickly attain and sustain over time very large expression levels, likely reflecting homeostatic mechanisms. We hypothesize that cells use negative feedback to prevent excessive transcription of one gene that could starve others, ensuring the efficient use of limited transcriptional and translational resources such as polymerases and ribosomes.

This finding, observed in bulk tissue data, was also confirmed at the sc level. Notably, with the greater resolution of sc data, we could even observe how the distribution of kinetic parameters evolves over time. By stratifying cell types based on their time of appearance, we observed that genes expressed in later-appearing cell types display fewer and more limited kinetic combinations, and we hypothesize that these cell types may adopt more constrained strategies to activate gene expression in a shorter amount of time. Overall, our findings reveal that not only are gene expression programs highly cell type-specific—determining which genes are expressed in a given cell type—

but also the kinetics of gene expression, dictating *how quickly* genes become expressed, are strongly cell type-specific.

Building on the observation that early cell types can "decide" which genes to express first, we investigated additional factors that influence the kinetics of gene expression. Of note, we discovered that essential genes—those involved in core cellular processes and critical for cellular fitness[41]—exhibit faster activation kinetics compared to non-essential genes. This difference was particularly pronounced in early-appearing cell types, where essential genes displayed higher kinetic rate constants ($k$ parameter) and were closer to their saturation levels. Traditionally, gene essentiality has been assessed through genetic perturbation screenings that evaluate the viability of mutant cells. However, there is a growing movement toward more quantitative approaches that account for broader aspects of a gene's contribution to cellular function beyond viability[41]. Our findings provide an alternative perspective on quantifying gene essentiality by incorporating the kinetics of gene activation into its definition.

To explore the transcriptome-epigenome interplay, we first examined the types of epigenetic regulation (enhancer and silencer cCREs) associated with different genes. Our analysis revealed that genes regulated by diverse types of cCREs (both increasing and decreasing trends, termed poly-pattern genes) show larger changes in gene expression and are more likely to be involved in brain-specific functions compared to genes regulated by simpler (mono-pattern) regulomes. Next, we investigated whether gene expression could be predicted based on chromatin accessibility at associated cCREs over time. To avoid predefined assumptions about their temporal dynamics —such as whether changes in chromatin accessibility precede or follow changes in gene expression[16]—we developed a biRNN-based neural network architecture. This architecture refines predictions by incorporating information from both directions in time: anterior (chromatin changes preceding gene expression) and posterior (chromatin changes following gene expression). In addition, unlike previous approaches that primarily focused on modeling enhancer-like regulatory effects[54], our methodology predicts gene expression over time by accounting for both enhancer and silencer regulatory influences. Our modeling framework is especially valuable given that generating time-series multi-omics data is both costly and time-consuming. Beyond demonstrating that gene expression can be predicted over time from chromatin accessibility, our approach offers a practical alternative: users can perform time-series experiments using a single modality (e.g., ATAC-seq or DNase-seq) and apply our models to infer gene expression kinetics without the need for RNA-seq data. Specifically, users can cross-reference their lists of cCREs with publicly available enhancer and silencer annotations in the human and mouse genomes[42,55] to select the most appropriate model—enhancer or silencer—based on the regulatory elements associated with their genes of interest.

To further enhance the interpretability of our model, we applied the SHAP method and found that cCREs located closer to their target genes had greater predictive significance. This result suggests that proximal cCREs may play a more biologically meaningful role in gene regulation. On the other hand, we also observed that distant cCREs contributed more prominently to predict gene expression at intermediate time points, indicating that their additive effects are critical in shaping the kinetic patterns of gene expression.

Although the usage of the logistic ODE has proven effective for fitting monotonic signals, our ODE-fitting approach is limited in its ability to capture peak-shaped signals, ensuring regular profiles. To overcome this limitation, we developed a simplified method that merges two sigmoid functions to fit the small subset of peak-shaped signals. We further demonstrate that this approach closely approximates results obtained from B-spline fitting, while offering the added advantage of being more interpretable. A potential future direction involves fitting a sum of logistic functions, which can provide a better global fit, especially in the peak region, but the parameter meaning becomes more entangled. Additionally, to improve peak fitting, we suggest leveraging local minima and maxima. Finally, Gaussian distributions may be employed, as their limiting behavior is well-suited for capturing spike-like signal patterns often associated with transcriptional bursts, which are biologically relevant.

Our results emphasize the need to model gene regulation dynamically rather than statically. As future steps, chronODE could help capture the temporal order of molecular events, such as determining the sequence of TF binding to regulatory regions. Moreover, as time-series data become more accessible, our framework could enable differential kinetic analysis, identifying differences in gene regulation kinetics across conditions. This approach could offer deeper insights into disease progression and help link drug response kinetics to target gene regulation. Similarly, insights into how cell type specificity and gene essentiality shape gene expression kinetics can help identify targets for gene therapy and assess their effects on cellular function across tissues and cell types. For example, our results suggest that targeting genes expressed in late-appearing cell types during brain development via gene therapy may require similar types of interventions, since these genes are all likely to show the same type of expression kinetics. In contrast, genes expressed in early cell types may require more diversified strategies. As kinetic approaches begin to illuminate molecular mechanisms of drug resistance in cancer[56–58], we envision these transcriptional and epigenetic kinetic maps could play a critical role in the future for developing personalized therapeutic strategies. Overall, we anticipate that, in the long term, our computational framework and findings will find multidisciplinary applications beyond genomics, particularly in fields like pharmacogenetics and pharmacokinetics.

## Methods

### Bulk and single-cell multi-omics datasets from mouse brain

**Bulk RNA-seq data from three brain regions.** We analyzed maps of gene expression (polyA+ RNA-seq) generated by the ENCODE Consortium[27] from three mouse fetal brain regions (forebrain, midbrain, and hindbrain) across eight developmental time points (PC days 10.5, 11.5, 12.5, 13.5, 14.5, 15.5, 16.5, and the first post-natal [PN] day). In the subsequent numerical calculations, we chose to represent the PN time point as PC day 21, since the standard length of a mouse pregnancy is typically in the range 19–21 days[59].

We obtained gene expression matrices of transcript per million (TPM) values from the ENCODE portal for two biological replicates per region and time point (mouse genome assembly mm10; Gencode annotation version M21[60]; https://www.encodeproject.org/matrix/?type=Experiment&status=released&related_series.%40type=OrganismDevelopmentSeries&replicates.library.biosample.organism.scientific_name=Mus+musculus&life_stage_age=embryonic+10.5+days&life_stage_age=embryonic+11.5+days&life_stage_age=embryonic+12.5+days&life_stage_age=embryonic+13.5+days&life_stage_age=embryonic+14.5+days&life_stage_age=embryonic+15.5+days&life_stage_age=embryonic+16.5+days&life_stage_age=postnatal+0+days&biosample_ontology.term_name=forebrain&biosample_ontology.term_name=hindbrain&biosample_ontology.term_name=midbrain&assay_title=polyA+plus+RNA-seq). The corresponding experiments and files' accession numbers are also listed in Supplementary Table 1. In all our analyses we used $\log_2$-transformed TPM values (pseudocount of 1).

**Bulk DNase- and ATAC-seq data from three brain regions.** We downloaded DNase-seq and ATAC-seq maps[28] from the ENCODE portal (https://www.encodeproject.org/matrix/?type=Experiment&status=released&related_series.%40type=OrganismDevelopmentSeries&replicates.library.biosample.organism.scientific_name=Mus+musculus&assay_title=ATAC-seq&life_stage_age=embryonic+10.5+days&life_stage_age=embryonic+11.5+days&life_stage_age=embryonic+12.5+days

&life_stage_age=embryonic+13.5+days&life_stage_age=embryonic+14.5+days&life_stage_age=embryonic+15.5+days&life_stage_age=embryonic+16.5+days&life_stage_age=postnatal+0+days&biosample_ontology.term_name=forebrain&biosample_ontology.term_name=hindbrain&biosample_ontology.term_name=midbrain&assay_title=DNase-seq). Specifically, DNase-seq maps were available for time points 10.5, 11.5, 14.5, and PN, while ATAC-seq maps were available for time points 11.5, 12.5, 13.5, 14.5, 15.5, 16.5, and PN. The corresponding experiments and files' accession numbers are also listed in Supplementary Tables 2 and 3.

Next, we downloaded the catalog of ENCODE4 cCREs[42] for the mouse genome (accession number ENCSR412JPD, [https://www.encodeproject.org/annotations/ENCSR412JPD]), which comprises 926,843 cCREs. Using this catalog, we constructed a matrix of chromatin accessibility signals for each cCRE across the eight time points and the three brain regions. Given that ATAC-seq data were only available from PC day 11.5 onward, in order to maximize the number of time points shared between the chromatin accessibility and gene expression maps, we integrated DNase-seq and ATAC-seq data in a single time course, as follows. First, we identified active cCREs during the time course. Specifically, for each of the three regions, we downloaded bigBed files of ATAC-seq pseudo-replicated narrow peaks available for each time point. Using the BEDTools[61] (version 2.30.0) function `intersectBed`, we identified 282,907 cCREs with ATAC-seq peaks in at least one time point. In the case of DNase-seq, pseudo-replicated peaks were unavailable, so we instead used the BEDTools function `multiIntersectBed` to identify the peaks shared across all replicates for each time point. 316,549 cCREs reported DNase-seq peaks in at least one time point. We defined our set of 405,554 "active" cCREs as those that reported a DNase-seq and/or ATAC-seq peak in at least one time point. The list of bigBed files employed in this step is available in Supplementary Tables 2 and 3. Second, we built a time-course matrix of chromatin accessibility for active cCREs. We downloaded ATAC-seq bigWig files (fold change over control; two replicates per time point and region; Supplementary Table 3) and computed the average signal in the cCRE coordinates' window at each time point and replicate using the bigWigAverageOverBed tool. This yielded a $405,554 \times 7$ matrix for each replicate and region. We followed the same procedure for the DNase-seq signal (read-depth normalized signal; Supplementary Table 2), and obtained a $405,554 \times 4$ matrix for each replicate and region. Finally, to harmonize ATAC- and DNase-seq signal matrices, for each brain region we performed a joint quantile normalization across replicates and time points using the R package preprocessCore[62].

**Single-cell RNA-seq data from 41 brain cell types.** We analyzed publicly available scRNA-seq data from mouse embryonic development published by ref. 29. Specifically, we downloaded the processed time-series gene expression matrices from https://omg.gs.washington.edu/jax/public/download.html (see "gene expression across time-points"). These data include gene expression profiles for 190 cell types across 43 time points, ranging from embryonic day (E) 8.5 to PN day 0. Our analysis focused on brain-associated cell types, specifically those classified in the original paper under the clusters "Neuroectoderm and glia," "Intermediate neuronal progenitors," "CNS neurons," and "Oligodendrocytes". To determine the timing of cell type emergence, we used marker gene lists from the original publication. For each cell type, we identified (1) the earliest time point at which any gene was detected as expressed (i.e., expression > 0) and (2) the first time point at which its marker gene was observed to be expressed. We selected 41 cell types for which these two time points coincided, ensuring high confidence in our determination of cell type appearance.

### The chronODE framework
Below we describe the three main components of the chronODE framework (Supplementary Fig. 1).

### Step 1: Data preprocessing
**Batch correction.** We performed batch correction to account for unwanted effects due to experimental differences in gene expression across replicates. To this end, we used the R package Limma[63] (function `removeBatchEffect`), specifying the time course (encoded as PC days) as the design, and the biological replicate as the batch level. For the chromatin accessibility datasets, we performed batch correction to remove also unwanted effects due to experimental differences between the two assays (i.e., ATAC- and DNase-seq). Thus, we used the shared time points 11.5, 14.5, and PN between ATAC- and DNase-seq matrices to calibrate the differences between the two assays, and specified both the replicate and assay features as batch levels.

**Shift to positive range.** Batch correction can sometimes result in negative values, which are inconsistent with the inherently positive nature of biological signals like gene expression and chromatin accessibility. To address this, we applied a global shift to ensure all values in the matrices were positive. Specifically, for each brain region, we identified the minimum expression value across all genes and, if this value was negative, we added its absolute value to each gene, shifting the entire matrix upward while maintaining the relative distribution of values. We applied the same approach to cCRE chromatin accessibility signals.

**Time-series differential analysis.** To identify, in a given brain region, genes with significant changes in expression over time, we used the R package maSigPro[64]. Using the function `make.design.matrix()`, we constructed two different design matrices for linear (`degree = 1`) and quadratic (`degree = 2`) regression models, with replicates handled internally. We then applied the `p.vector()` function to each design matrix with the following parameters: `Q = 0.05`, `MT.adjust = "BH"`, `min.obs = 5`. We defined dynamic genes as those reporting a maSigPro Benjamini–Hochberg (BH) adjusted $p$ value < 0.01 in at least one of the two designs. We followed the same procedure to identify, in each brain region, cCREs with significant changes in chromatin accessibility over time.

### Step 2: Kinetic analysis
**Monotonic fitting.** Consider a vector $\mathbf{z}$ containing time-series measurements, where each element represents the genomic signal $z$ at a given time point. The minimum and maximum signals in $\mathbf{z}$ are defined as $z_{min}$ and $z_{max}$, respectively (see also Supplementary Fig. 3F). In this study, we analyzed two types of genomic signals $z$: the level of expression of a given gene, and the level of chromatin accessibility of a given cCRE. To model these signals, we used a "simplified" form of the logistic ODE (Eq. 2, Supplementary Fig. 2). For more details, see Supplementary Note 1 (Propositions 1 and 6). Below, we outline the data normalization, fitting, and inverse transformation procedures involved in the monotonic fitting step.

*Data normalization.* We propose normalizing the data to ranges that align with the theoretical framework of the simplified form of the logistic ODE (Eq. 2). Specifically, prior to the ODE fitting step, we normalize each gene/cCRE vector $\mathbf{z}$ to two different ranges ([$10^{-5}$, 1] and [1, 2]), and subsequently select the fit with the lowest MSE (see next section, "ODE fitting"). Note that time $t$ is invariant to this normalization step.

- **Data normalization to the [$10^{-5}$, 1] range.** The min-max normalization is applied to obtain a new vector $\mathbf{y}^*$ such that the lower bound ($y^*_{min}$) approaches 0 (for more details, see Supplementary Note 1, Proposition 6):

$$\mathbf{y}^* = \frac{(\mathbf{z} - z_{min})(R_{max} - R_{min})}{z_{max} - z_{min}} + R_{min} \tag{9}$$

where $R_{min} = 10^{-5}$ and $R_{max} = 1$.

This range is suitable, for instance, to model signals that exhibit an increasing pattern with an acceleration component.

- **Data normalization to the [1, 2] range**: To accommodate signals primarily matching the deceleration phase of the logistic curve (where an earlier acceleration phase would lie in the negative space), we also considered normalizing **z** to the [1, 2] range. This is achieved by shifting the normalized values in the $[10^{-5}, 1]$ range upward by one unit. This adjustment approaches the lower bound with $\frac{b^*}{2}$ and the upper bound with $k^*$, connecting the signal to the deceleration phase of the logistic curve.

Assuming a lower asymptote of zero in the normalized range (i.e., $a^* = 0$), we demonstrate that the kinetic parameters $k^*$ and $b^*$ can be directly related to the data in the original (observed) range (i.e., **z**) (see section below "Restoring the fitted curve to the original range of the data").

*ODE fitting*. Upon data normalization, our pipeline performs ODE fitting of the gene/cCRE **y*** vector through the following steps:

- **Estimation of the kinetic parameters $k^*$ and $b^*$**. This step requires initial guesses for $k^*$ and $b^*$. For $b^*$, we set an initial guess of 1.5, whereas for $k^*$ we considered both positive and negative initial guesses ($\pm 0.9$). We fit the simplified ODE using each one of the following four combinations of input parameters: ($+ k^*, b^*, $ **y*** in range $[10^{-5}, 1]$); ($- k^*, b^*, $ **y*** in range $[10^{-5}, 1]$); ($+ k^*, b^*, $ **y*** in range $[1, 2]$); ($- k^*, b^*, $ **y*** in range $[1, 2]$). In each of these four combinations, $t_{\text{start}}$ is defined as the first time point monitored by the study (for instance, when applying chronODE on the bulk RNA-seq data from forebrain, $t_{\text{start}} = 10.5$), while $y^*_{\text{start}}$ is defined as the expression level of a given gene in the forebrain at day 10.5 following the normalization step (either in the range $[10^{-5}, 1]$ or in the range $[1, 2]$ for the first and last two combinations, respectively). The fitting is performed using the Scipy functions `curve_fit` (maximum calling number equal to `5000`) and `odeint`, with the latter using LSODA, an adaptive steps algorithm[65]. In our pipeline, we repurposed the solver in such a way that it returns an NA solution if it cannot find optimized parameters in the predefined logistic parameter space.

- **Reconstruction of the smooth fitted line**. For each combination of input parameters that converged on a positive $b^*$ solution, we reconstructed the corresponding smooth fitted line ($\mathbf{y}^*_{\text{fitted}}$) using the analytical solution of the simplified form of the logistic ODE (i.e., Eq. 3).

- **MSE calculation and choice of best fitting solution**. For all converged solutions within both $[10^{-5}, 1]$ and $[1, 2]$ ranges, we computed the MSE between the corresponding $\mathbf{y}^*$ and $\mathbf{y}^*_{\text{fitted}}$ vectors, and selected the $\mathbf{y}^*_{\text{fitted}}$ solution with the lowest MSE.

*Restoring the fitted curve to the original range of the data*. To preserve the biological interpretation of $\mathbf{y}^*$ signals, we apply an inverse transformation of the fitted parameters $a^*$ and $b^*$ to obtain $a$ and $b$, respectively, using the following formula (see Supplementary Note 1, Proposition 6, and Supplementary Fig. 2):

$$a = \frac{(a^* - R_{\text{min}})(z_{\text{max}} - z_{\text{min}})}{R_{\text{max}} - R_{\text{min}}} + z_{\text{min}}$$
$$b = \frac{(b^* - R_{\text{min}})(z_{\text{max}} - z_{\text{min}})}{R_{\text{max}} - R_{\text{min}}} + z_{\text{min}} \tag{10}$$

where $R_{\text{min}}$ and $R_{\text{max}}$ represent the lower and upper bounds of the range to which the data was previously normalized ($[R_{\text{min}}, R_{\text{max}}]$). For example, $R_{\text{min}} = 10^{-5}$ and $R_{\text{max}} = 1$ for data normalized in the $[10^{-5}, 1]$ range, while $R_{\text{min}} = 1$ and $R_{\text{max}} = 2$ for data normalized in the $[1, 2]$ range. Notably, the kinetic parameter $k$ remains unchanged in the normalized and original ranges (i.e., $k^* = k$; see Supplementary Note 1, Propositions 7 and 8).

With the original $z_{\text{start}}$, and $a$ and $b$ determined, we use the analytical solution of the generalized logistic ODE (Equation 1) to recover the fitted curve in the original data range, denoted as $z(t)$ (see Supplementary Note 1, Proposition 5):

$$z(t) = \frac{(b - a)Ce^{kt}}{1 + Ce^{kt}} + a. \tag{11}$$

Finally, we note that the position of the inflection (switching) point over time is preserved under normalization (for more details, see Supplementary Note 1, Proposition 9). Specifically, the second derivative of the rescaled curve remains proportional to that of the originally fitted curve, ensuring that the inflection point and the related kinetic patterns are maintained. This ensures that the biological and mathematical interpretation of the results remains accurate and consistent within the context of the original data range.

We implemented this process as a fully automated Nextflow[66] (version 23.04.1) pipeline to enable genome-wide analysis. Specifically, the pipeline takes as input an $n \times m$ matrix, where $n$ represents the number of elements (e.g., genes or cCREs) as rows, and $m$ represents the numeric time points (e.g., the eight time points in mouse PC days) as columns. Each cell in this matrix corresponds to the gene expression or cCRE chromatin accessibility signal for a given element at a specific time point.

The pipeline produces two output matrices. The first output matrix contains the reconstructed smooth fitted values in the original range across the $m$ time points for each element. The second output matrix contains the fitted parameters $k$ and $b$ for each element, as well as information on the kinetic class to which genes/cCREs belong to (see Methods section "Kinetic classification"). This robust implementation ensures scalability and reproducibility for large-scale genomic datasets.

**Gaussian mixed modeling.** Gaussian mixed modeling (GMM) has been previously used for anomaly detection tasks[67]. Based on this, here we used a GMM-based approach to classify genes and cCREs as "acceptable" or "unacceptable" fits based on the distribution of their MSEs. Accordingly, we model the MSE distribution by two Gaussian components: one representing acceptable fits with low MSE values, and another representing unacceptable fits with high MSE values (Supplementary Fig. 1), assigning each data point (gene or cCRE) a probability of belonging to each group. A cutoff threshold is then determined to distinguish between the two clusters. Data points with MSE values exceeding this threshold were classified as unacceptable fits.

Specifically, for a given dataset containing either genes' or cCREs' ODE-fitted genomic signals restored to the original range, we utilized the function `normalmixEM()` from the R package mixtools[68] (setting parameter `k=2`) to fit the GMM, and the function `plot_cut_point()` from the R package plotGMM to calculate the cutoff threshold for acceptable MSE values. To showcase this approach in the case of monotonic fits for bulk RNA-seq data, in Supplementary Fig. 3D, E we reported the GMM-partitioned MSE distributions for the three brain regions as well as examples of genes with acceptable and unacceptable fits.

**Piecewise fitting.** For those cases with unacceptable monotonic fits, we next applied the logistic function in a piecewise manner, identifying the global maximum or minimum point of a given gene or cCRE signal. First, the left segment is fitted. To ensure zero-order continuity, the logistic function is adjusted so that the endpoint of the fitted left segment aligns with the starting point of the right segment. To achieve first-derivative continuity, we apply quadratic B-spline (degree of 2) on the fitted data to ensure profile regularity. As the MSE between the ODE-based and the B-spline-based fittings approaches 0

(Supplementary Fig. 4F), indicating that the two lines coincide, this approach ensures continuity and is kinetically interpretable thanks to the presence of the $k$ and $b$ parameters. The growth/decay rate constant was computed as the average absolute $k$ value ($k_{avg}$) between the left and right segments, denoted as $k_{left}$ and $k_{right}$, respectively. To evaluate the quality of the fit, we computed the GMM threshold based on the MSE values obtained from the piecewise fitting, following the same methodology described for the monotonic fits.

**Kinetic classification.** Focusing on the lists of cCREs and genes with acceptable monotonic fits, we next computed $t_{switch}$, $t_{minimum}$, and $t_{saturation}$ based on the fitted parameters $k$ and $b$ as well as the initial $y^*_{start} = y^*(t_{start})$ value, employing the following formulas (see Supplementary Note 1, Propositions 3 and 4 for a description of these formulas):

$$t_{switch} = \frac{1}{k^*} ln\left(\frac{b^*}{y^*_{start}} - 1\right) + t_{start} \qquad (12)$$

$$t_{minimum} = \frac{1}{k^*} ln\left(\frac{10^{-16} b^*}{y^*_{start}}\right) + t_{start} \qquad (13)$$

$$t_{saturation} = \frac{1}{k^*} ln\left(99 \frac{b^*}{y^*_{start}} - 99\right) + t_{start} \qquad (14)$$

The catalogs of $t_{switch}$, $t_{minimum}$, and $t_{saturation}$ for genes (expression) and cCREs (chromatin accessibility) in the three mouse brain regions are available in Supplementary Data 1.

This approach enabled us to classify genes and cCREs into three main classes based on their estimated $t_{switch}$: switchers ($t_{switch}$ occurs between days 10.5 and PN), accelerators ($t_{switch}$ occurs after day PN), and decelerators ($t_{switch}$ occurs before day 10.5) (Supplementary Fig. 5A and Supplementary Data 1).

**Step 3: Predicting gene expression from cCRE chromatin accessibility using a bidirectional Neural Network.** To predict gene expression over time, we used the time-series vector of cCRE chromatin accessibility signals as input to a bidirectional RNN (biRNN). Across the three brain regions, more than 99% of monotonic genes had fewer than 60 associated cCREs. Therefore, for each gene, we selected the 60 closest cCREs from its associated set (for further details on this aspect see next section "Mono-pattern and poly-pattern genes"). We implemented the biRNN with the following architecture: `Input_size = 60` (i.e., $m = 60$ in Fig. 5A), `Hidden_layer_size = 30`, and `ReLu(Dense(10, 1))` (i.e., $n = 10$ in Fig. 5A). We note that the dimensions of the hidden layers of the biRNN and the dense layer serve as hyperparameters, providing flexibility to encode the input. Each of the chromatin signals, at each time point, serves as an input for the network (Fig. 5A). The input is defined as a 3D tensor where the first position stands for the number of samples, the second position is the number of cCRE-related-features, and the third position is an 8-dimensional vector of the different epigenetic signals over time. The output is also a 3D tensor, where the first position points to the same number of samples as the input, each corresponding to one gene across the 8 time points. We used PyTorch[69] to train the network and set the batch size equal to 4, along with an MSE as the loss function and an initial learning rate of 0.0001 using the Adam optimizer. The network was trained with 200 epochs. We combined data from the three brain regions and further split the gene-cCRE pairs into two groups depending on whether the chromatin accessibility was positively or negatively correlated with the gene expression. Next, each of these two groups was divided into two further subgroups (i.e., mono- and poly-pattern genes), resulting in four distinct datasets for training. We trained and tested the model separately on these four groups in a ratio

80:20. Throughout the training process we computed the MSE for each sample $i$ in a batch ($MSE_i$), which captures the average squared differences between the true and predicted output values across the eight time points:

$$MSE_i = \frac{1}{8} \sum_{t=1}^{8} \left(g_{true,i,t} - g_{pred,i,t}\right)^2. \qquad (15)$$

Next, we minimized the overall loss value by averaging across the entire batch:

$$Loss = \frac{1}{N} \sum_{t=1}^{N} (MSE_i), \qquad (16)$$

where $N$ is the batch size.

## Mono-pattern and poly-pattern genes

We focused on genes that reported acceptable monotonic fits for gene expression and linked them to cCREs that also reported acceptable monotonic fits for chromatin accessibility. The linking strategy was based on the degree of proximity between cCREs and genes. Specifically, we assigned to every cCRE the closest gene based on linear distance, using the BEDTools `closest` utility.

Next, we grouped cCREs into two types of regulatory trends: activated ($k > 0$) and repressed ($k < 0$). We used the UpSetR package[70] to compute how many genes are linked to these different combinations of cCRE regulatory trends (Supplementary Fig. 7A). We defined as mono-pattern (mono-p.) those genes that are associated either with one cCRE or with multiple cCREs that all share the same regulatory trend (Fig. 4A). We defined as poly-pattern (poly-p.) those genes associated with multiple cCREs that have both increasing and decreasing trends.

To functionally characterize mono-pattern and poly-pattern genes, we performed gene ontology (GO) enrichment analysis on both sets of genes. In this specific case, we focused on two core sets of genes that were consistently classified across the three regions as mono-pattern and poly-pattern. For each of the two sets of genes, we used the GOStats R library[71] and a gene universe of protein-coding genes from the Gencode M21 annotation. We then selected significantly enriched Biological Process (BP) terms using a BH-adjusted $p$ value cutoff of 0.01 (Fig. 4D).

Given that genes linked to multiple cCREs are more likely to exhibit poly-pattern regulation, we sought to explore the independent contributions of the number of cCREs and the type of regulatory trend (mono-pattern vs. poly-pattern) towards a gene's likelihood of being associated with specific GO terms. In particular, we focused on the top 15 GO terms most significantly enriched among poly-pattern genes, many of which are related to brain development (Fig. 4D). For this analysis, we focused again on the two core sets of mono-pattern and poly-pattern genes used as input to the previous GO analysis. For a given GO term, we assigned a binary variable $y$ to each gene, where $y = 0$ indicates no association with the GO term and $y = 1$ indicates an association. The association of a gene to the GO term was determined through a mapping object generated with the R package org.Mm.eg.db, where each GO ID (key) maps to a set of Entrez Gene IDs (values) associated with that GO term in the mouse genome. We then modeled the set of $y$ values from poly-pattern and mono-pattern genes as a response variable in the following logistic regression model:

$$y = ax + bz + c(x:z) \qquad (17)$$

where $x$ is the average number of cCREs associated with the gene across the three brain regions, $z$ is a binary variable indicating whether the gene is mono-pattern ($z = 0$) or poly-pattern ($z = 1$), and $x{:}z$ represents the interaction between $x$ and $z$. We used the R function `glm`

with the `family = binomial(link = "logit")` option to fit the model and retrieve coefficients $a$, $b$, and $c$, as well as the associated $p$ values (Supplementary Fig. 7B). For each of the three types of coefficients, we obtained BH-adjusted $p$ values across the 15 GO terms. To rule out biases in our results due to multicollinearity between independent variables, we used the R package car (function `vif`) to estimate variance inflation, obtaining in all tested cases a VIF value $< 1.5$. We performed a similar analysis to evaluate the contribution of $x$ and $z$ towards a gene's dynamic fold-change (Fig. 4B). In this case $y$ is defined as the expression fold-change of a given gene, calculated as the difference between maximum and minimum expression level of the gene during the time course, and it serves as response variable of a linear model that we fit with the R function `lm`. The results of these analyses are shown in Supplementary Fig. 7C.

## Quantification and statistical analysis

All statistical analyses were performed using the R (version 4.2.3) or Python (version 3.9.16) languages, as specified in the Methods and/or figure legends. Unless otherwise specified, plots were made with the R package ggplot2[72] or the Python package matplotlib[73]. All box plots depict the first and third quartiles as the lower and upper bounds of the box, with a band inside the box showing the median value and whiskers representing $1.5 \times$ the interquartile range.

## Reporting summary

Further information on research design is available in the Nature Portfolio Reporting Summary linked to this article.

## Data availability

All the data employed in this study are publicly available. ENCODE data were obtained from the ENCODE Portal (www.encodeproject.org) with the accession numbers provided in Supplementary Tables 1–3. The catalog of mouse cCREs has been downloaded from the ENCODE portal with the accession number ENCSR412JPD[42]. The single cell RNA-seq data have been obtained from https://omg.gs.washington.edu/jax/public/download.html (see "gene expression across timepoints")[29]. The processed data resulting from the kinetic classification of genes and cCREs across the three brain regions are available in Supplementary Data 1. Source data are provided with this paper.

## Code availability

Code associated with the chronODE framework is publicly available at https://github.com/gersteinlab/chronODE[74].

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

## Acknowledgements

This work was supported by the Albert L. Williams Professorship at Yale University and by funding from the National Human Genome Research Institute (grant U01 HG013840), awarded to M.G. We thank Edoardo Borsari (Università di Bologna) for discussions regarding the mathematical proofs presented in this study.

## Author contributions

B.B. and M.F. conceived the project and designed the study. M.F., K.X., and B.B. contributed to mathematical proofs supporting the computational framework. M.F. developed the biRNN modeling framework. B.B., M.F., E.S.W., K.X., and S.X.L. performed the computational analyses. X.Y. contributed tools and ideas to perform computational analyses. M.G. acquired funding, supervised the study, and provided scientific feedback. B.B. and M.F. wrote the manuscript with the contribution of all authors. E.S.W., K.X., and S.X.L. contributed equally to this work.

## Competing interests

The authors declare no competing interests.
