## [Transparent Peer Review file · Nature Communications]

The chronODE framework for modelling multi-omic time series with ordinary differential equations and machine learning

Corresponding Author: Professor Mark Gerstein

Version 0:

Reviewer comments:

Reviewer #1

(Remarks to the Author)

In this study, the authors developed an innovative computational method called chronODE that directly models chromatin and gene-expression kinetics from time-resolved measurements, based on ordinary differential equations. The model used is a simple yet intuitive two-parameter (k, b) logistic function that is capable of describing a wide range of monotonic “switching” behavior. The authors showed that chronODE captures cell type-specific kinetics of chromatin accessibility, that poly-pattern regulated genes during brain development are more dynamic in gene expression and more enriched in brain-specific functions than mono-pattern regulated genes, and that the modeled rate of change in chromatin accessibility can be used to predict the rate of change in target gene expression over time with reasonable accuracy. Overall, this is an important and well-written study that provides a unified and versatile framework to jointly model chromatin and gene-expression kinetics from multi-omics time-series data.

While the authors provide adequate justification for the “deceleration” phase of the logistic model in terms of the carry capacity or saturation driven by the number of nucleosomes present at a given genomic locus, the “acceleration” phase of the logistic model is not as well justified. It is possible to think of other models that only have the deceleration/saturation phase, but not the acceleration phase. For example, the logarithm function, or a linear increase followed by saturation. Why is the acceleration phase necessary? The authors discussed this in more detail in the Discussion section, but this needs to be better justified when the logistic model is first introduced.

The authors used the chronODE modeling results to classify cis-regulatory elements into three categories: accelerator, switcher, and decelerator. The authors showed that these classifications already provide a lot of insights into the dynamics of brain development. I am wondering if additional insights can also be gained by analyzing the logistic function parameters (k, b) from these chronODE modeling results? For example, how are (k, b) parameters distributed for all cis-regulatory regions during brain development? Is there anything interesting that can be learned from the (k, b) parameter distributions?

The authors presented a very exciting result that poly-pattern regulated genes are more dynamic in gene expression and more enriched in brain-specific functions than mono-pattern regulated genes. However, one possible confounding factor is the number of cis-regulatory regions associated with the target gene, since genes regulated by many cis-regulatory regions are probably more likely to be poly-pattern than genes regulated by few cis-regulatory regions, everything else being equal. The authors need to make sure that their results and conclusions are not affected by this possible confounding factor.

The authors used neural networks and random forests to show that the modeled rate of change in chromatin accessibility can be used to predict the rate of change in target gene expression over time with reasonable accuracy. What about the performance of a base-line linear model? If the linear model performs much worse, then the authors can argue that the inclusion of non-linear effects is essential here.

(Remarks on code availability)

Reviewer #2

(Remarks to the Author)

Borsari and colleagues present a simple, but efficient approach for the dynamical modeling of gene expression regulation based on time-resolved data. To describe chromatin accessibility, they used DNase-seq and ATAC-seq data from the ENCODE registry of mouse candidate cis regulatory elements (cCREs). The mathematical model is simply a logistic function, where they vary or estimate its only two parameters k and b . In order to use the model for classification of data, they also consider the inflection time point (called t_{switch}) and whether that is reached during the measurement. Accordingly, they subdivide the cCREs into switchers, accelerators or decelerators. In addition, they consider some cCREs as stable, i.e. not showing temporal dynamics in their accessibility. Based on that simple classification, they present different types of statistics for mouse brain samples at different developmental stages and from different brain regions. Next, they also apply the approach to human fetal brain development data, giving percentages of the different classes for, e.g. different neuron types or fetal-specific or adult-specific cCREs.

To extend the approach to gene expression, they link all the cCREs to their nearest gene, which results in assignment of one (mono-) or more (poly-pattern) cCREs to genes. They again present some basic statistics for the distribution of a now slightly shifted classification (early/late increasing and decreasing instead of switchers, accelerators and decelerators) among mono-, bi- and poly pattern genes. Finally, they fit an unspecified model linking gene expression data for mono-pattern genes to the model of their associated cCRE, using neural networks or random forest. The extension to poly-pattern and the nature of the linking function remain unsolved.

Overall, this is an interesting approach to have a basic quantitative and time-resolved description linking chromatin accessibility and gene expression. It is substantiated with large consortium-based data sets.

I have a few points.

1. I appreciate the attempt to use a simple modeling approach to derive a generally applicable description of the process. I find it, however, a bit puzzling, to push a new branding of the classical logistic function and now try to call it chronODE. I didn't find an explanation for that word creation and also no scientific justification.
2. Several cases follow, where it is claimed that "chronODE numerically solves ...", "chronODE estimated...", what I think was done by the authors with the help of their python programs.
3. Another naming issue: over generations it has made sense to discriminate temporal changes of location in space (measured as velocity) from temporal changes in state (measured) as rates. Here, both are mixed ad libitum. But I assume, it is not considered how cCREs or genes physically move, their states are relevant.
4. Very importantly, while the authors have presented many nice statistical plots of their classification results, I fail to see what the newly gained biological insight is, beyond ENABLING direct comparison and biological interpretation of the kinetics. The discussion refers to regulatory elements reaching full accessibility or inaccessibility at the end of time course – can that not also directly be seen from the data? The authors could compare and elaborate on the additional value of having the model included.
5. The link between cCRE accessibility and gene expression has been only established for monopattern cases. The authors argue that it can be extended to multiple cCRE chromatin features or multiple cCREs (polypattern). This remains to be shown since the assignment of the specific contributions of more features or more cCREs to the expression of a single gene appears to be a far more complex problem than the presented one-to-one assignment.
6. In Line 258 pp. it is claimed that "accurately predict" expression changes of target genes over time. Though I appreciate that it is a prediction and maybe even valuable, I can't find the evidence for the accuracy. Figure 5b shows a modest correlation and Figure 5c selected examples of genes, where the reader doesn't get any idea about how well it works for the rest. One might think: these are the three genes where it worked.... Given the complexity of biological regulation, I would even not expect accurate prediction from just one feature. Could the authors elaborate on that?

Minor points

Line 44: "velocity of all genes" rate of expression change of all genes

Line 123: define "early" and "late"

Line 131: k_i is a rate constant, not a rate. It should also not be interpreted as such.

(Remarks on code availability)

Reviewer #3

(Remarks to the Author)

Borsari and co-workers present a mathematical framework based on ordinary differential equations (ODEs) and machine/deep learning to analysis time-stipulated epigenetic and gene expression datasets. Although the topic is timely and such framework would be very useful to the community for investigating abnormal development, diseased tissue or perturbed conditions (e.g. gene KO, drug exposure), the proposed modeling framework suffers from limitations and questionable choices and the documentation of its assessment using real data is incomplete, making it difficult to draw any conclusions on the validity of the approach. Here are my detailed comments:

About the ODE modeling of cCREs:

-The first step of data interpolation before fitting the ODE model is very unusual and should not be pursued as it inflates the model-to-data error, dilutes the importance of real data points among synthetic ones, and consequently messes up the MSE computations. In other words, such preliminary step transforms the input data in a way that worsens the model fit to true data. I strongly suggest removing it as ODE model calibration can be performed directly on the datapoints, as widely done by the

community.

-The reason why cCRE data is normalized is unclear. The actual absolute value of the signal may contain information which is lost through this normalization step. And this information can be integrated in the framework by estimating y_0 actual value, (possibly after a rescaling to ensure that all data points are positive). This will bring additional value to the framework which could offer to analyze not only relative cCREs values as it is the case now, but also absolute time-profiles, thus allowing for comparison between cCREs. Importantly, this may allow to use the ODE logistic model as an input in the machine/deep learning models (see comment below).

-The ODE fitting assessment is not properly reported which makes it impossible to assess the validity of the model, and of the choice of the logistic function. The authors only report MSE values which are relative to a given problem and do not provide the information of what is a good or a bad fit. We need to see the MSE distribution over all cCRE and examples of what is acceptable/unacceptable. The authors states that they excluded the 20% of elements with the highest reported MSE but this threshold is not justified. The subsequent question is about the reason of the bad fits and whether they support to introduce a more complex model with additional degrees of freedom in your model. Indeed, rejected cCRE time-profiles from data analysis because they look more complex than a logistic curve is not a biologically-sound argument, and the framework must handle such cases. In other words, the modeling needs to adapt to the data, and not the other way round.

About the machine/deep learning models:

- it is unclear why the authors chose to use the derivatives of cCRE (dC_i/dt) as inputs instead of directly using the amount of CRE (C_i). Indeed, corresponding ODE models would have used linear (e.g. law of mass action) or non-linear (e.g. Hill functions) terms involving C_i , and not dC_i/dt . This choice adds the complexity of estimating dC_i/dt which is not directly documented in the data and needs to be approximated in an additional step that is associated with additional error (see also next comment).

-For estimating the derivatives of cCRE amount (dC_i/dt), the authors have used linear interpolation, which is a bad choice, as it propagates the experimental error (see my comment above). Considering that the first part of this study is dedicated to the design of an ODE model that precisely provides the derivative of C_i , it is surprising that the authors did not use it as input of their deep learning regressors. Indeed, contrarily to linear interpolation, such ODE-based model tends to generate smoother profiles, that may even correct for part of the experimental error when present in only one data point of time-profile, for instance.

-Performances of the proposed RF or NN models are low as indicated by low correlation between model and data. More importantly, the models create peaks or troughs in the time-profiles which were not present in the data, thus impairing the subsequent biological interpretation of the data (Fig5c, sup. Fig. 7). As they are now, such unreliable algorithms may not be used by the community.

-The proposed machine/deep learning models are rather simple whereas more complex model structure should be employed here to properly predict gene time profiles. Indeed, the authors claims that the specificity of the framework lays in the analysis of time-stipulated data whereas the RF or NN models do not account for this time structure of the data, which may explain the large discrepancies observed between the true and predicted gene profiles. Authors should consider using NN framework like neural ODEs that are likely to give better results for this specific problem.

-Assuming that the expression of a gene may be predicted by a single cCRE is a questionable choice. It should probably be relaxed to increase model performance.

-RF models are not properly trained as hyperparameters (e.g. number of trees) should have been determined by cross-validation.

(Remarks on code availability)

Version 1:

Reviewer comments:

Reviewer #1

(Remarks to the Author)

The authors have adequately addressed all my comments. I recommend publication of the manuscript in Nature Communications.

(Remarks on code availability)

Reviewer #3

(Remarks to the Author)

The authors have responded to most of my concerns. Please find below some remaining questions.

Major points:

- Related to Ref3.2: The current model only partly recapitulates the data as the absolute values of the gene levels are not present in the k and b parameters. The information is in the R_{min} and R_{max} scaling factors and the minimum the authors could do is to present statistics of these factors. The fact that the authors need to normalize the data to fit them suggests that the model is not flexible enough and would benefit from the addition of an extra parameter allowing for the modification of the value of the lower asymptote of the curve.
- Model fit: the authors need to provide the values of (t_0, y_0) they have used in the fit and their justification.
- 1595: what are those y_0 values? Where do they come from since they are not estimated?
- Ref3.3b More details would be appreciated regarding the piecewise fit: how do the authors decide on the time cut-off between the logistic curves? Do they optimize it in the fit? Are the logistic curves fitted sequentially or simultaneously. To ensure the profile regularity, a constraint on the first derivative should be applied at the time cut-off. Regarding goodness of fit, do they use the same threshold for MSE as for monotonic profiles?
- l. 619-621: "We [...] further split the gene-cCRE pairs into two groups depending on whether the chromatin accessibility was positively or negatively correlated with the gene expression." This information implies that the gene data is available. Hence, one can question the usefulness of the deep learning model that precisely aims to predict the gene profile from cCRE data.

Minor revisions:

- L.80 confusing formulation: "We numerically solve Equation 1 by optimizing the kinetic parameters k and b to reconstruct y , ": You have solved analytically the ODE to get the equation, and you need to estimate the parameter to evaluate numerically the explicit formulation of y .
- l. 80, please refer to supplementary note, Proposition 1, for derivation of analytical formula.
- The authors need to clarify that, with their formulation, $y=0$ is an asymptote of the curve, meaning that it is never reached, except when y starts at 0. Figure 1 and several sentences (e.g. figure 1 legend) are misleading and should be clarified. Similarly, it should be stated that y_0 needs to be different from 0.
- Supp Note, Proof of proposition 1: Change the notation of c at first mention as you have not demonstrated that it is equal to the one in Proposition 1 yet. Next, you need to treat the case when $y>b$ or include the constraints $y<b$ in the proposition.
- Equation 14: there is probably a typo as it should read the sum of squareroot of MSE, (and not square) to obtain the typical L2 norm. If this is not a typo, this would require more explanation as this would be an unusual choice of Loss.

(Remarks on code availability)

Reviewer 1

Ref 1.0. - Reviewer's summary

Reviewer's comment	In this study, the authors developed an innovative computational method called chronODE that directly models chromatin and gene-expression kinetics from time-resolved measurements, based on ordinary differential equations. The model used is a simple yet intuitive two-parameter (k, b) logistic function that is capable of describing a wide range of monotonic "switching" behavior. The authors showed that chronODE captures cell type-specific kinetics of chromatin accessibility, that poly-pattern regulated genes during brain development are more dynamic in gene expression and more enriched in brain-specific functions than mono-pattern regulated genes, and that the modeled rate of change in chromatin accessibility can be used to predict the rate of change in target gene expression over time with reasonable accuracy. Overall, this is an important and well-written study that provides a unified and versatile framework to jointly model chromatin and gene-expression kinetics from multi-omics time-series data.
Author's response	We thank the reviewer for their positive feedback on our manuscript.

Ref 1.1. - Justifying the acceleration phase of the logistic curve

Reviewer's comment	While the authors provide adequate justification for the "deceleration" phase of the logistic model in terms of the carry capacity or saturation driven by the number of nucleosomes present at a given genomic locus, the "acceleration" phase of the logistic model is not as well justified. It is possible to think of other models that only have the deceleration/saturation phase, but not the acceleration phase. For example, the logarithm function, or a linear increase followed by saturation. Why is the acceleration phase necessary? The authors discussed this in more detail in the Discussion section, but this needs to be better justified when the logistic model is first introduced.
Author's response	We thank the reviewer for this comment and agree that the limited description of the acceleration phase in our previous version of the manuscript may have implied it is not an intrinsic feature of the data. As requested, we have now added in our revised manuscript a more detailed justification of the acceleration phase both in the introduction and in the results section where the logistic function is first introduced (see excerpts below). Briefly, we describe that, in the case of chromatin accessibility and histone modifications, the acceleration phase of the logistic curve

	represents the cooperativity aspect that characterizes nucleosome remodelling and histone modifications. For instance, as reported by Kim et al., (2022, Cell Rep), 'DNA unwraps more easily [from the nucleosome] the more unwrapped it already is.' This 'easier unwrapping' corresponds to the acceleration phase of the logistic curve, indicating that DNA unwrapping occurs at an increasingly rapid rate over time. In the context of gene expression, the acceleration phase mimics the rapid rise in mRNA copies following transcriptional bursting, as described by Golding et al., (2005, Cell). Furthermore, the acceleration phase supports gradual changes, which are biologically/physically meaningful, compared to a case where there is abrupt transition from slope m (linear increase) followed by saturation (slope 0). Finally, we would like to emphasize that the acceleration phase is frequently observed in our data, either as part of the full logistic curve in switcher signals, or as a dominant, "stand-alone" pattern in a smaller subset of cases (accelerators). In summary, the usage of the logistic curve to model our data is well justified, as it not only fits the data but also provides strong biological and physical relevance compared to other curves that might fit mathematically but lack meaningful biological interpretation. For instance, we chose not to use the logarithmic function which is composed of both negative and positive y values, as our database contains biological signals, which are inherently positive by nature. We admit that there are other switching functions we could have used, such as \tanh, but the logistic is an intuitive switcher function. Furthermore, the usage of the logistic function is aligned with other recent studies, such as Ma et al., (2020, Cell) who have also reported logistic-like behavior in their single-cell SHARE-seq datasets (chromatin accessibility and gene expression).
Excerpt from manuscript	Excerpt from main manuscript (Introduction): "Epigenetic processes, such as chromatin accessibility and histone modifications, are shaped by two fundamental properties: cooperativity and saturation, both governed by fundamental physical and chemical constraints, related to the characteristics and quantities of nucleosomes, transcription factors (TFs), and other molecular components. Cooperativity refers to the self-reinforcing nature of molecular interactions. For instance, when chromatin starts to open, DNA tends to unwrap more easily from nucleosomes the more unwrapped it already is¹⁰. Similarly, once deposited at a specific locus, post-translational modifications of histones (e.g., methylation or acetylation) can rapidly propagate to neighboring histones¹¹⁻¹³. On the other hand, these processes are characterized by natural limits, causing them to reach saturation over time. [...] Notably, these properties also apply to gene expression kinetics. [...]" Excerpt from main manuscript (Results): "Given the cooperative and saturating nature of genomic signals (Fig. 1), we propose modeling transcriptomic and epigenomic kinetics over time using the logistic function [...]. We describe the logistic curve as consisting of two biologically meaningful phases. First, the acceleration phase (where y' progressively increases over time) models the cooperative aspect of epigenetic and transcriptional processes, which tend to occur more frequently and easily after the initial state^{10-14,16}. Second, the deceleration phase (where y' progressively decreases) models the saturating aspect of these processes (Fig. 1)." Excerpt from new Figure 1:

Ref 1.2 - Interpreting k and b parameters

Reviewer's comment	The authors used the chronODE modeling results to classify cis-regulatory elements into three categories: accelerator, switcher, and decelerator. The authors showed that these classifications already provide a lot of insights into the dynamics of brain development. I am wondering if additional insights can also be gained by analyzing the logistic function parameters (k, b) from these chronODE modeling results? For example, how are (k, b) parameters distributed for all cis-regulatory regions during brain development? Is there anything interesting that can be learned from the (k, b) parameter distributions?
Author's response	We thank the reviewer for this insightful suggestion. As requested, we have now added a new plot of the distribution of the k and b parameters across all genes and analyzed them in greater detail (see below excerpts of results and discussion sections, and new Figure 2A-B, new Figure 3A-B, and new Supplementary Figure 3C-E). Briefly, when analyzing the joint distribution of k and b parameters across genes, we observed that these two parameters follow an L-shaped distribution, with genes displaying three major combinations of k-b values: high k & low b (Q1), low k & low b (Q2), and low k & high b (Q3) (Figure 2A-B). Specifically, genes with fast expression kinetics (high k) tend to have low saturation levels (low b), while genes with slow expression kinetics (low k) have a less constrained saturation level b, which can be either low or high. As

	we write in the discussion, we hypothesize that this is due to biophysical limitations: when the saturation level (b) is large, it is biologically unlikely to achieve high expression (b) while maintaining rapid expression kinetics (high k) because the cell's transcriptional and translational machinery—such as polymerases and ribosomes—cannot support both a high b and a high k simultaneously. As a result, we hypothesize that cells may limit the expression of rapidly transcribed genes via negative feedback mechanisms, preventing their b values from being too high. We extended this analysis also to time-series single-cell RNA-seq data and observed a similar L-shaped distribution of k and b parameters across most cell types (see excerpt from new Figure 3A-B below). Stratifying cell types by their time of emergence revealed that later-emerging cell types exhibited k values concentrated at higher levels, which we interpret as reflecting the need for these cell types to rapidly ramp up gene expression. This shift in the distribution of k and b parameters corresponded to changes in the kinetic classification of genes. Specifically, genes expressed in early cell types displayed a broader range of k/b combinations and were closer to saturation, aligning with decelerator profiles. In contrast, genes expressed in late-emerging cell types predominantly exhibited two main k vs. b combinations, which were strongly negatively correlated, consistent with switcher or accelerator profiles. These findings underscore the critical influence of cell type emergence timing on gene expression kinetics.
Excerpt from manuscript	Excerpt from main manuscript (Results): “When analyzing the distribution of k and b values across genes and brain regions, we found that the monotonic genes are clustered into three major quadrants based on their k-b combinations: high k & low b ($Q1$), low k & low b ($Q2$), and low k & high b ($Q3$) (Fig. 2A). This distribution exhibits an L-shaped pattern (Pearson's r correlation = -0.44, p value $< 2.2e-16$), where high values of both k and b are not observed together (i.e., $Q4$ is absent). [...] (Fig. 2B) We performed a similar analysis for genes with peak-like expression profiles, and found that, also in this case, only a small fraction ($\sim 8\%$) corresponds to $Q4$ genes (Supplementary Fig. 3C-E). Altogether, this suggests that genes cannot undergo rapid expression changes while simultaneously achieving high saturation. Consequently, the expression level of rapidly saturating genes is fundamentally constrained ($Q1$ genes). [...] Genes with distinct kinetic behaviors, as defined by the three quadrants in Fig. 2A, were observed also across most individual cell types, consistent with the diversity of gene expression kinetics identified in the bulk tissue analysis. However, the L-shaped distribution of these three kinetic subpopulations changed markedly over time as new cell types emerged [...] (Fig. 3A). Excerpt from main manuscript (Discussion):

“To demonstrate its utility, we applied *chronODE* to bulk and *sc* multi-omic datasets from mouse brain development. The ability to fit the expression profiles of most genes with a logistic function suggests that gene expression is a smooth process characterized by gradual changes over time. Moreover, the observation that logistic parameters exhibit an L-shaped distribution emphasizes that gene expression is fundamentally constrained, similar to other chemical and biological phenomena^{50,51}. In particular, we found that it is almost impossible for genes to quickly attain and sustain over time very large expression levels, likely reflecting homeostatic mechanisms. We hypothesize that cells use negative feedback to prevent excessive transcription of one gene that could starve others, ensuring the efficient use of limited transcriptional and translational resources such as polymerases and ribosomes.”

Excerpt from new **Figure 2A-B**:

Figure 2: Kinetic characterization of the developing mouse brain transcriptome across three brain regions. **A:** Distribution of the kinetic parameters k (magnitude expressed in absolute value, y axis) and b (x axis) across all monotonic genes in the three brain regions. Genes are grouped into three quadrants (Q1–Q3) based on k -means clustering of their k - b combinations. No genes are found in Q4. **B:** Lineplot showing, for each set of activated and repressed genes in Q1 through Q3 groups, the average expression (y axis) over time (x axis; PC = Post-Conception). Note that the average expression levels (expressed in \log_2 (TPMs)) were rescaled to the range 0-100% to allow for comparison across Q1-Q3 groups. Q1 activated and repressed genes have an average expression profile that resembles the full logistic curve, whereas Q2 and Q3 genes align with the decelerator and accelerator parts of the curve, respectively. The arrow in Q3 genes indicates that these profiles remain far from reaching the saturation level b .

Excerpt from new **Figure 3A-B**:

Figure 3

Figure 3: Cell type specificity of gene expression kinetics. **A:** Distribution of the kinetic parameters k (y axis) and b (x axis) for activated genes ($k > 0$) across 41 brain cell types. Cell types have been grouped based on their day of appearance (between 8-9, 9-10, 10-11, 12-13, and 14-15 embryonic [E] days). **B:** For each cell type (x axis) within these five groups, proportion of genes (%) (y axis) that are characterized by switcher (purple), decelerator (green), and accelerator (orange) profiles.

Excerpt from new **Supplementary Figure 3C-E:**

Supplementary Figure 3: Properties of Q1-Q4 genes. **C:** Distribution of the kinetic parameters k (magnitude expressed in absolute value, y axis) and b (x axis) across all genes with piecewise fits in the three brain regions. For genes with piecewise sigmoid fits, two k values are computed (k_{left} and k_{right} one for each sigmoid segment) and here we display their average absolute k value (k_{avg}). The plot subdivision in four Q1-Q4 quadrants is the same shown in Figure 2A. **D-E:** Lineplot showing, for genes modelled by piecewise fits in Q1 through Q4, the average expression (y axis)

	over time (x axis; PC = Post-Conception). Expression profiles with concave peaks are characterized by $k_{left} > 0$ (D), while those with convex peaks have $k_{left} < 0$ (E). Note that the average gene expression levels (expressed in \log_2 TPMs) were rescaled to the range 0-100% to allow for comparison across Q1-Q4 groups.
--	---

Ref 1.3 - Potential confounding factor in mono- vs. poly-pattern genes analysis

Reviewer's comment	The authors presented a very exciting result that poly-pattern regulated genes are more dynamic in gene expression and more enriched in brain-specific functions than mono-pattern regulated genes. However, one possible confounding factor is the number of cis-regulatory regions associated with the target gene, since genes regulated by many cis-regulatory regions are probably more likely to be poly-pattern than genes regulated by few cis-regulatory regions, everything else being equal. The authors need to make sure that their results and conclusions are not affected by this possible confounding factor.
Author's response	We thank the reviewer for their positive feedback on these results. The reviewer is actually making a very good point about whether our results on the type of cCRE regulation (mono/polypattern) are confounded by the number of cCREs. As we explain below, we have tested this hypothesis and found that the reviewer is correct. Specifically, both the number of cCREs and the type of regulation (mono-/poly-pattern) are significantly associated with the dynamic range of expression and the brain-specific function of poly-pattern genes. We have featured these new analyses in the results section and in new Supplementary Figure 6B-C (see excerpts below). Briefly, to test this hypothesis, for every GO term enriched either among poly-pattern genes (Figure 4D), we have performed a logistic regression model using the following formula: $y = ax + bz + c(x:z)$ where y is the probability of the gene to be associated with a given GO term (0 = no association; 1 = association); x is the number of cCREs associated with the gene; z is a binary variable indicating whether the gene is mono-pattern or poly-pattern (0 = mono-pattern; 1 = poly-pattern); $x:z$ is the interaction term between x and z. Doing so, we tested the significance of the a, b, and c coefficients. We found that for most of the brain-related GO terms enriched among poly-pattern genes, all three factors x, z, and $x:z$ reported coefficients significantly different from 0 (Supplementary Figure 6B). This indicates that both the number and the type of cCREs (poly-pattern or mono-pattern) are significantly associated with the type of function performed by the gene. We have obtained similar results when testing the association between gene

	expression fold-change (dependent variable y) and the number and type of cCREs (x, z, and $x:z$) (Supplementary Figure 6C). Also, note that to strengthen this analysis, we have simplified the definition of poly and mono-pattern genes by considering only two major trends of chromatin accessibility, either increasing ($k > 0$) or decreasing ($k < 0$), to better connect these results to the ODE fitting section (see new Figure 4A). Therefore, in the revised version of the manuscript we have converted bi-pattern genes either to mono-pattern (if the bi-patterning previously involved early and late increasing cCREs) or poly-pattern (if it previously involved decreasing and early/late increasing).
Excerpt from manuscript	Excerpt from Methods: “Given that genes linked to multiple cCREs are more likely to exhibit poly-pattern regulation, we sought to explore the independent contributions of the number of cCREs and the regulatory pattern type (mono-pattern vs. poly-pattern) towards a gene’s likelihood of being associated with specific GO terms [...]. For this analysis we focused again on the two core sets of mono-pattern and poly-pattern genes used as input to the previous GO analysis. For a given GO term, we assigned a binary variable y to each gene, where $y = 0$ indicates no association with the GO term and $y = 1$ indicates an association. The association of a gene to the GO term was determined through a mapping object generated with the R package org.Mm.eg.db, where each GO ID (key) maps to a set of Entrez Gene IDs (values) associated with that GO term in the mouse genome. We then modelled the set of y values from poly-pattern and mono-pattern genes as a response variable in the following logistic regression model: $y = ax + bz + c(x:z) \quad (15)$ where x is the average number of cCREs associated with the gene, z is a binary variable indicating whether the gene is mono-pattern ($z = 0$) or poly-pattern ($z = 1$), and $x:z$ represents the interaction between x and z. We used the R function <code>glm</code> with the family = binomial(link = "logit") option to fit the model and retrieve coefficients a, b, and c, as well as the associated p values (Supplementary Figure 6B). For each of the three types of coefficients, we obtained BH-adjusted p values across the 15 GO terms. We performed a similar analysis to evaluate the contribution of x and z towards a gene’s dynamic fold-change [...]. The results of these analyses are shown in Supplementary Figure 6C.” Excerpt from main manuscript (Results): “A central question is what drives the highly dynamic behavior and specialized functions of poly-pattern genes: is it the number of regulatory elements or the diversity of their regulatory trends? To address this, we disentangled the contributions of cCRE count and trend diversity (mono- vs. poly-pattern regulation). Our analysis demonstrated that both the number of cCREs and the diversity of their trends significantly correlate with a gene’s dynamic expression profile and its involvement in brain-specific functions (Supplementary Fig. 6B-C). This finding expands on our conclusion that gene

expression is fundamentally a constrained process by having a balanced regulation involving both activation and repression mechanisms.”

Excerpt from new **Supplementary Figure 6B-C**:

Supplementary Figure 6: Properties of mono- and poly-pattern genes. **B:** Logistic regression analysis of gene-GO term associations. The left heatmap shows the $-\log_{10}$ Benjamini-Hochberg adjusted p values for the cCRE count coefficient, the poly-pattern coefficient, and their interaction across the 15 GO terms most significantly associated with poly-pattern genes. The right heatmap displays the corresponding regression coefficients for each term. **C:** Analogous representation for the results obtained through a linear model estimating the effect of cCRE count and poly-pattern regulation (independent variables) on gene expression fold-change (dependent variable).

Excerpt from new **Figure 4A**:

Figure 4

Figure 4: Genes linked to poly-pattern cCREs are more dynamic during brain development and are enriched in brain-specific functions. A: Schematic illustrating the distinction between mono-pattern and poly-pattern genes. Mono-pattern genes (left panel) can be linked to one or more cCREs, but if associated with multiple cCREs, all of these cCREs exhibit the same regulatory profile—either all activated ($k > 0$, red, upper panel) or all repressed ($k < 0$, blue, lower panel). Genes linked to a single cCRE represent the simplest case of mono-pattern regulation. In contrast, poly-pattern genes (right panel) are associated with multiple cCREs, which may be predominantly activated (upper panel) or repressed (lower panel).

Ref 1.4 - Linear vs. non-linear model

Reviewer's comment	The authors used neural networks and random forests to show that the modeled rate of change in chromatin accessibility can be used to predict the rate of change in target gene expression over time with reasonable accuracy. What about the performance of a base-line linear model? If the linear model performs much worse, then the authors can argue that the inclusion of non-linear effects is essential here.
Author's response	We thank the reviewer for this suggestion. As requested, we have compared our results to those obtained with a base-line linear model and described the benefit of including non-linear effects in the prediction (see excerpt below for new Supplementary Figure 7D). Briefly, compared to a base-line linear NN (i.e., a NN where we removed bidirectional RNN and ReLU operations), our non-linear NN reported an improvement up to 15.55% in true-predicted mean correlations (Supplementary Figure 7D). We found that the bidirectional RNN (biRNN) especially improves the prediction for poly-pattern genes. On the other hand, the ReLU activation function ensures predictions to be in the range of positive values, aligning with the characteristics of gene expression values. Finally, we note we have removed the RF results since they did not provide more meaningful information compared to the NN model.
Excerpt from manuscript	Excerpt from new Supplementary Figure 7D:  Supplementary Figure 7: biRNN predictions across four gene regulatory mechanisms. D: Distributions of cross-timepoint Pearson's correlations between true and predicted gene expression values using a linear (orange) or non-linear (green) model. The baseline linear model corresponds to a neural network (NN) without bidirectional RNN (biRNN) and ReLU activation. Statistically significant differences between distributions were assessed using the Wilcoxon paired test: ns = not significant (p value > 0.05), * = p value < 0.05, ** = p value < 0.01. For poly-pattern genes, the linear NN shows significantly lower correlations than the

non-linear NN (enhancer poly-pattern: p value = 0.0043; silencer poly-pattern: p value = 0.0285). No significant differences were observed for mono-pattern genes (enhancer mono-pattern: p value = 0.736; silencer mono-pattern: p value = 0.2227).

References for responses to Reviewer 1

Kim, J., Sheu, K. M., Cheng, Q. J., Hoffmann, A. & Enciso, G. Stochastic models of nucleosome dynamics reveal regulatory rules of stimulus-induced epigenome remodeling. *Cell reports* **40** (2022).

Golding, I., Paulsson, J., Zawilski, S. M. & Cox, E. C. Real-time kinetics of gene activity in individual bacteria. *Cell* **123**, 1025–1036 (2005).

Ma, S. et al. Chromatin Potential Identified by Shared Single-Cell Profiling of RNA and Chromatin. *Cell* **183**, 1103–1116 (2020).

Reviewer 2

Ref 2.0.a - Reviewer's summary

Reviewer's comment	Borsari and colleagues present a simple, but efficient approach for the dynamical modeling of gene expression regulation based on time-resolved data. To describe chromatin accessibility, they used DNase-seq and ATAC-seq data from the ENCODE registry of mouse candidate cis regulatory elements (cCREs). The mathematical model is simply a logistic function, where they vary or estimate its only two parameters k and b. In order to use the model for classification of data, they also consider the inflection time point (called t_{switch}) and whether that is reached during the measurement. Accordingly, they subdivide the cCREs into switchers, accelerators or decelerators. In addition, they consider some cCREs as stable, i.e. not showing temporal dynamics in their accessibility. Based on that simple classification, they present different types of statistics for mouse brain samples at different developmental stages and from different brain regions. Next, they also apply the approach to human fetal brain development data, giving percentages of the different classes for, e.g. different neuron types or fetal-specific or adult-specific cCREs. To extent the approach to gene expression, they link all the cCREs to their nearest gene, which results in assignment of one (mono-) or more (poly-pattern) cCREs to genes. They again present some basic statistics for the distribution of a now slightly shifted classification (early/late increasing and decreasing instead of switchers, accelerators and decelerators) among mono-, bi- and poly pattern genes. Overall, this is an interesting approach to have a basic quantitative and time-resolved description linking chromatin accessibility and gene expression. It is substantiated with large consortium-based data sets. I have a few points.
Author's response	We thank the reviewer for their constructive feedback. In the revised version of our manuscript we have done extensive work to address the limitations that the reviewer has correctly pointed out, as detailed in the response to the following comments.

Ref 2.0.b - General points to be addressed

Reviewer's comment	Finally, they fit an unspecified model lining gene expression data for mono-pattern genes to the model of their associated cCRE, using neural networks or random forest. The extension to poly-pattern and the nature of the linking function remain unsolved.
--

Author's response	We thank the reviewer for this comment. First, we acknowledge that our initial approach of using two models—the Random Forest (RF) and Neural Network (NN)—to predict gene expression from chromatin signals at associated cCREs may have been unclear, as noted by the reviewer's description of it as "unspecified." To address this, we have streamlined our methodology in the revised manuscript by focusing solely on the NN and have removed the results related to the RF. Second, the reviewer suggested extending the predictive model to include poly-pattern genes. In response, we now present an enhanced version of our NN model in the revised manuscript, capable of predicting time-series expression for both mono- and poly-pattern genes. Specifically, the model accounts for four regulatory scenarios of gene activation and repression: mono-pattern enhancers, mono-pattern silencers, poly-pattern enhancers, and poly-pattern silencers. Predictions are generated by linking gene expression at each time point to a linear combination of cCREs. Following this linear combination, nonlinear activation functions (e.g., tanh and ReLU) are applied. This revised approach is described in detail in the manuscript (see excerpt below).
Excerpt from manuscript	Excerpt from main manuscript (Results): "Building on these findings and utilizing our many-to-one linking schema—where multiple cCREs are linked to a given gene over time—we designed a model architecture (Fig. 5A) capturing four types of gene regulatory mechanisms that differ in the direction (enhancer/silencer) and diversity (mono-p./poly-p.) of cCRE regulation as follows: 1) mono-pattern genes subject to enhancer regulation ("enhancer mono-p."), where all cCRE signals linked to a gene are positively correlated with its expression; 2) mono-pattern genes subject to silencer regulation ("silencer mono-p."), where all cCRE signals linked to the gene are negatively correlated with its expression; 3) poly-pattern genes subject to enhancer regulation, where the majority of cCRE signals are positively correlated with gene expression, while a minor fraction exhibit silencer-like effects ("enhancer poly-p."); and 4) poly-pattern genes subject to silencer regulation, where most cCRE signals are negatively correlated with gene expression, with a smaller proportion displaying enhancer-like effects ("silencer poly-p."). This approach enabled us to train our model towards predicting both the direction and kinetic patterns of gene expression over time. Each subset of genes was used to train the model independently, with an 80:20 split for training and testing, respectively." Excerpt from new Figure 5A:

Ref 2.1 - chronODE naming issue

Reviewer's comment	I appreciate the attempt to use a simple modeling approach to derive a generally applicable description of the process. I find it, however, a bit puzzling, to push a new branding of the classical logistic function and now try to call it chronODE. I didn't find an explanation for that word creation and also no scientific justification. Several cases follow, where it is claimed that "chronODE numerically solves ...", "chronODE estimated ...", what I think was done by the authors with the help of their python programs.
Author's response	We thank the reviewer for this comment and we realize our previous wording might have seemed a bit confusing. To address this concern, we have edited the text in both the Results and the Discussion sections of the manuscript to make it clearer that we use the logistic solver embedded within the chronODE framework to solve the ODEs. We have also emphasized that the novelty of chronODE lies in that it is a software wrapping the steps of ODE fitting into one single command and enabling the application of this approach genome-wide. To make this point clearer, we have cited examples of other widely used software that wrap established statistical models or equations to analyze genomics data, such as DESeq2 (R package that uses Generalized Linear Models (GLMs) to perform differential gene expression; Love et al., 2014 Genome Biol), or PLINK (Chang et al., 2015 Gigascience), a largely used

	tool to perform Genome-Wide Association Studies based on linear and logistic regression. We hope now it is clearer that our aim is not to rebrand the classical logistic function but rather to propose a useful piece of software to analyze and quantify the kinetics of time-series multi-omics data and classify genes and regulatory elements based on their kinetic patterns.
Excerpt from manuscript	Excerpt from main manuscript (Results): “Specifically, our proposed chronODE package optimizes this solution by repurposing a generic initial value problem solver and constraining the numerical solution to the logistic space, so that parameters k and b can be easily interpreted.” Excerpt from main manuscript (Discussion): “To facilitate the application of this ODE at the genome-wide scale, we developed chronODE, a software that integrates the ODE-fitting workflow into a single command, similar to other genomic tools that package established models into user-friendly software^{46,47}.”

Ref 2.2 - Definition of velocity

Reviewer’s comment	Another naming issue: over generations it has made sense to discriminate temporal changes of location in space (measured as velocity) from temporal changes in state (measured) as rates. Here, both are mixed ad libitum. But I assume, it is not considered how cCREs or genes physically move, their states are relevant.
Author’s response	We agree with the reviewer that the term “velocity” is not the most appropriate choice since it usually describes a temporal change of location in space. However, in the biological community there is a vast literature on this topic using the term velocity to describe gene expression changes as defined by a scalar and a direction component (La Manno et al., 2018 Nature; Bergen et al., 2020 Nat Biotechnol; Chen et al., 2022 Sci Adv; Li et al., 2023 Nat Biotechnol). In any case, in the revised version of our manuscript we do not use anymore the derivatives (previously referred to as velocities) of cCRE/gene signals for our predictions, but rather the actual values of gene and cCRE signals.

Ref 2.3 - Biological insights gained through our framework

Reviewer’s comment	Very importantly, while the authors have presented many nice statistical plots of their classification results, I fail to see what the newly gained biological insight is, beyond ENABLING direct comparison and biological interpretation of the kinetics. The discussion refers to regulatory elements reaching full accessibility or inaccessibility at the end of time course – can that not also directly be seen from the data? The authors could
---

	compare and elaborate on the additional value of having the model included.
Author's response	We agree with the reviewer that our previous wording in the Discussion did not sufficiently emphasize the additional value of our framework and the biological insights we can gain through it. As requested, in the revised version of the manuscript we have now greatly expanded the discussion of new biological insights (see excerpt below). First of all, directly comparing time-series genomic signals is challenging due to noise in chromatin accessibility and gene expression data, which often obscures major trends. Our approach smooths profiles and quantifies temporal kinetics of genes and cCREs, enabling an objective classification that reveals patterns not easily detectable otherwise. This information is crucial for applications like gene therapy, where understanding the timing of gene activity can guide interventions: for instance, if we plan to silence the expression of a given gene, it is important to do so before the mRNA is accumulated in considerable amounts. Consequently, it is important to precisely estimate when this accumulation occurs. Second, our framework enabled us to gain several biological insights: 1) we found that both at the tissue and at the level of individual cell types, gene expression is characterized by gradual changes, likely in response to homeostatic principles (see new Figure 2A-B and new Figure 3A-B); 2) we observed that gene expression kinetics are shaped by different factors, including cell-type specificity, gene essentiality, and regulatory mechanisms (Figure 3C-D and Supplementary Figure 7B). In particular, we found that late appearing cell types show more constrained kinetic patterns of gene activation, compared to early cell types. Furthermore, we show that early appearing cell types prioritize the expression of essential genes by enabling their faster activation kinetics. As a result, essential genes tend to reach their maximum level of expression earlier in time compared to non-essential genes.
Excerpt from manuscript	Excerpt from main manuscript (Discussion): “Time-series functional genomic assays are a powerful way to investigate the kinetics of activation and repression of multiple genes and regulatory elements. However, characterizing these kinetics and linking them to biological processes over time poses significant challenges. Both epigenomic and transcriptomic datasets are inherently noisy and characterized by diverse kinetic patterns, making it difficult to discern major trends by simply comparing raw data. We showed that by using a logistic ordinary differential equation (ODE) we can model multiple kinetic patterns at the genome-wide scale, and we find that these kinetics are associated with different cell types emerging over time. Specifically, this ODE effectively captures the cooperative and saturation-like behaviors of genomic signals, such as nucleosome remodeling, post-translational modification of histones, and gene expression. To facilitate the application of this ODE at the

genome-wide scale, we developed *chronODE*, a software that integrates the ODE-fitting workflow into a single command, similar to other genomic tools that package established models into user-friendly software^{46,47}. For example, with *chronODE*, we can quantitatively estimate where a gene is in its expression life cycle during a given process—whether it is close to saturation or in the acceleration phase of its expression, a characteristic that cannot be precisely quantified from the raw data and requires a rigorous modelling approach.

This methodology offers several advantages. First, it uses a simple and intuitive mathematical function to model time-series genomic signals. Prior approaches to model this type of data have often relied on higher-order polynomial functions (e.g., B-splines), which are difficult to interpret because each gene is fitted with a unique set of base functions optimized to match data trends, lacking a direct connection with the underlying biology^{48,49}. In contrast, the logistic ODE yields two intuitive and interpretable kinetic parameters with direct biological meaning.

To demonstrate its utility, we applied *chronODE* to bulk and sc multi-omic datasets from mouse brain development. The ability to fit the expression profiles of most genes with a logistic function suggests that gene expression is a smooth process characterized by gradual changes over time. Moreover, the observation that logistic parameters exhibit an L-shaped distribution emphasizes that gene expression is fundamentally constrained, similar to other chemical and biological phenomena^{50,51}. In particular, we found that it is almost impossible for genes to quickly attain and sustain over time very large expression levels, likely reflecting homeostatic mechanisms. We hypothesize that cells use negative feedback to prevent excessive transcription of one gene that could starve others, ensuring the efficient use of limited transcriptional and translational resources such as polymerases and ribosomes.

This finding, observed in bulk tissue data, was also confirmed at the sc level. Notably, with the greater resolution of sc data, we could even observe how the distribution of kinetic parameters evolves over time. By stratifying cell types based on their time of appearance, we observed that genes expressed in later-appearing cell types display fewer and more limited kinetic combinations, and we hypothesize that these cell types may adopt more constrained strategies to activate gene expression in a shorter amount of time. Overall, our findings reveal that not only are gene expression programs highly cell type-specific—determining which genes are expressed in a given cell type—but also the kinetics of gene expression, dictating how quickly genes become expressed, are strongly cell type-specific.

Building on the observation that early cell types can 'decide' which genes to express first, we investigated additional factors that influence the kinetics of gene expression. Of note, we discovered that essential genes—those involved in core cellular processes and critical for cellular fitness³⁸—exhibit faster activation kinetics compared to non-essential genes. This difference was particularly pronounced in early-appearing cell types, where essential genes displayed higher kinetic rate constants (k parameter) and were closer to their saturation levels. Traditionally, gene essentiality has been assessed through genetic perturbation screenings that evaluate the viability of mutant cells. However, there is a growing movement toward more quantitative approaches that account for broader aspects of a gene's contribution to cellular function beyond viability³⁸. Our findings provide a new perspective on quantifying gene essentiality by incorporating the kinetics of gene activation into its definition.

To explore the transcriptome-epigenome interplay, we first examined the types of epigenetic regulation (enhancer and silencer cCREs) associated with different genes. Our analysis revealed that genes regulated by diverse types of cCREs (both increasing and decreasing trends, termed poly-pattern genes) show larger changes in gene expression and are more likely to be involved in brain-specific functions compared to genes regulated by simpler (mono-pattern) regulomes. Next, we investigated whether gene expression could be predicted based on chromatin accessibility at associated cCREs over time. To avoid predefined assumptions about their temporal dynamics—such as whether changes in chromatin accessibility precede or follow changes in gene expression¹⁶—we developed a biRNN-based neural network architecture. This architecture refines predictions by incorporating information from both directions in time: anterior (chromatin changes preceding gene expression) and posterior (chromatin changes following gene expression). In addition, unlike previous approaches that primarily focused on modeling enhancer-like regulatory effects⁵², our methodology predicts gene expression over time by accounting for both enhancer and silencer regulatory influences.

To further enhance the interpretability of our model, we applied the SHAP method and found that cCREs located closer to their target genes had greater predictive significance. This result suggests that proximal cCREs may play a more biologically meaningful role in gene regulation. On the other hand, we also observed that distant cCREs contributed more prominently to predict gene expression at intermediate time points, indicating that their additive effects are critical in shaping the kinetic patterns of gene expression.

Our results emphasize the need to model gene regulation dynamically rather than statically. As future steps, chronode could help capture the temporal order of molecular events, such as determining the sequence of TF binding to regulatory regions. Moreover, as time-series data become more accessible, our framework could enable differential kinetic analysis, identifying differences in gene regulation kinetics across conditions. This approach could offer deeper insights into disease progression and help link drug response kinetics to target gene regulation. Similarly, insights into how cell type specificity and gene essentiality shape gene expression kinetics can help identify targets for gene therapy and assess their effects on cellular function across tissues and cell types. For example, our results suggest that targeting genes expressed in late-appearing cell types during brain development via gene therapy may require similar types of interventions, since these genes are all likely to show the same type of expression kinetics. In contrast, genes expressed in early cell types may require more diversified strategies. As kinetic approaches begin to illuminate molecular mechanisms of drug resistance in cancer⁵³⁻⁵⁵, we envision these transcriptional and epigenetic kinetic maps could play a critical role in the future for developing personalized therapeutic strategies. Overall, we anticipate that, in the long term, our computational framework and findings will find multidisciplinary applications beyond genomics, particularly in fields like pharmacogenetics and pharmacokinetics.”

Excerpt from new **Figure 2A**:

Figure 2

Figure 2: Kinetic characterization of the developing mouse brain transcriptome across three brain regions. A: Distribution of the kinetic parameters k (magnitude expressed in absolute value, y axis) and b (x axis) across all monotonic genes in the three brain regions. Genes are grouped into three quadrants ($Q1$ – $Q3$) based on k -means clustering of their k - b combinations. No genes are found in $Q4$.

Excerpt from new **Figure 3:**

Figure 3

Figure 3: Cell type specificity of gene expression kinetics. *A*: Distribution of the kinetic parameters k (y axis) and b (x axis) for activated genes ($k > 0$) across 41 brain cell types. Cell types have been grouped based on their day of appearance (between 8-9, 9-10, 10-11, 12-13, and 14-15 embryonic [E] days). *B*: For each cell type (x axis) within these five groups, proportion of genes (%) (y axis) that are characterized by switcher (purple), decelerator (green), and accelerator (orange) profiles. *C*: Distribution of kinetic parameter k (y axis) for essential (red) and non-essential (gray) genes (x axis). *D*: Proportion (%) (y axis) of essential (“Ess.”) and non-essential (“Non-Ess.”) genes (x axis) belonging to the three kinetic classes (color-coded).

Excerpt from new **Supplementary Figure 7B**:

Ref 2.4 - Predicting the expression of poly-pattern genes

Reviewer's comment	The link between cCRE accessibility and gene expression has been only established for monopattern cases. The authors argue that it can be extended to multiple cCRE chromatin features or multiple cCREs (polypattern). This remains to be shown since the assignment of the specific contributions of more features or more cCREs to the expression of a single gene appears to be a far more complex problem than the presented one-to-one assignment.
Author's response	We agree with the reviewer that predicting gene expression based on a single cCRE represents a simplified scenario, whereas, in reality, genes are often regulated by multiple cCREs (Choi et al., 2021 eLife). As requested, in the revised manuscript we have expanded our approach and predicted the expression also of genes associated with multiple cCREs (either monopattern or polypattern). Overall, our updated comprehensive modeling framework now handles four distinct regulatory mechanisms: (1) mono-pattern genes associated with enhancer cCREs, (2) mono-pattern genes associated with silencer cCREs, (3) poly-pattern genes associated with predominantly enhancer cCREs, and (4) poly-pattern genes associated with predominantly silencer cCREs as described in the excerpt below.

Excerpt from manuscript

Excerpt from main manuscript (Results):

“Building on these findings and utilizing our many-to-one linking schema—where multiple cCREs are linked to a given gene over time—we designed a model architecture (Fig. 5A) capturing four types of gene regulatory mechanisms that differ in the direction (enhancer/silencer) and diversity (mono-p./poly-p.) of cCRE regulation as follows: 1) mono-pattern genes subject to enhancer regulation (“enhancer mono-p.”), where all cCRE signals linked to a gene are positively correlated with its expression; 2) mono-pattern genes subject to silencer regulation (“silencer mono-p.”), where all cCRE signals linked to the gene are negatively correlated with its expression; 3) poly-pattern genes subject to enhancer regulation, where the majority of cCRE signals are positively correlated with gene expression, while a minor fraction exhibit silencer-like effects (“enhancer poly-p.”); and 4) poly-pattern genes subject to silencer regulation, where most cCRE signals are negatively correlated with gene expression, with a smaller proportion displaying enhancer-like effects (“silencer poly-p.”). This approach enabled us to train our model towards predicting both the direction and kinetic patterns of gene expression over time. Each subset of genes was used to train the model independently, with an 80:20 split for training and testing, respectively.”

Excerpt from new Figure 5A:

Figure 5

Figure 5: Predicting time-series gene expression from chromatin signals of associated cCREs.
A: Architecture of the biRNN-based model. At each time point, the input for a specific multicRE–gene pair i is a vector \vec{c} of dimension $1 \times m$, where m represents the number of cCRE signals associated with the gene. The model generates an output vector, representing the gene expression levels \vec{g} across all time points.

Ref 2.5 - Clarification on the accuracy of predictions

Reviewer’s comment

In Line 258 pp. it is claimed that “accurately predict” expression changes of target genes over time. Though I appreciate that it is a prediction and maybe even valuable, I can’t find the evidence for the accuracy. Figure 5b shows a modest correlation and Figure 5c selected examples of

	genes, where the reader doesn't get any idea about how well it works for the rest. One might think: these are the three genes where it worked.... Given the complexity of biological regulation, I would even not expect accurate prediction from just one feature. Could the authors elaborate on that?
Author's response	We agree with the referee that the sentence “accurately predict” along with showing only three examples may seem to be overestimating the model's performance, especially given the previously reported correlations between true and predicted values. To respond, in the revised version of the manuscript, we have improved our model adopting a more conventional approach that predicts the actual values of gene expression rather than the derivatives, and this modification has led to a significant increase in the model performance. We evaluated the model's performance in two ways. First, we computed, for every gene, the correlation between true and predicted values across time points (cross-timepoint correlation). We achieved mean correlations between 0.57-0.90, which are higher than those previously reported in the literature. For instance, Zhou et al. 2017 Nat Commun, who developed a method for predicting chromatin accessibility from gene expression across different cell types, reported mean correlations in the range 0.35-0.39. Second, we computed the correlation between true and predicted values across all genes (cross-gene correlation) and reported values between 0.87-0.97. Again these values are higher compared to those reported by Zhou et al., which are in the range 0.65-0.82. We have now clarified this aspect in the text by specifying that the goodness prediction is relative to the state of the art. We also note that our intention is not to show the best three genes but to provide representative examples of our predictions. In our revised manuscript we show a larger number of examples.
Excerpt from manuscript	Excerpt from main manuscript: “To evaluate the performance of the model, we first computed cross-gene correlations between true and predicted expression values. Our predictions on the test set showed a strong positive correlation (>0.85) in all the four types of regulatory mechanisms (Fig. 5B), higher compared to previous reports of chromatin-gene expression predictions⁴⁴. We also evaluated the model performance by computing the distribution of Mean Squared Error (MSE) and cross-timepoint correlation between true and predicted gene expression values (Supplementary Fig. 7C). In particular, we achieved mean cross-timepoint correlations ranging between 0.57 and 0.90, higher than the equivalent cross-cell type correlations reported previously⁴⁴. To further showcase the predictive capability of our model, we also included some examples of true vs. predicted gene expression time-series profiles (Fig. 5C).” Excerpt from new Figure 5B-C:

Figure 5: Predicting time-series gene expression from chromatin signals of associated cCREs.

B: Density plot showing cross-gene expression correlation for each of the four regulatory mechanisms. Each scatterplot shows true (x axis) versus predicted (y axis) expression values (\log_2 -transformed TPMs) across all genes in the test set. **C:** Representative examples of gene expression predictions over time demonstrating our model's capability of capturing the three

kinetic patterns (switchers, decelerators, and accelerators). True and predicted expression values are color-coded. Expression values correspond to \log_2 -transformed TPMs.

Excerpt from new **Supplementary Figure 7C**:

Supplementary Figure 7: biRNN Predictions across four gene regulatory mechanisms. C. Left panel: Cross-timepoints correlation computed between the true and predicted values for each gene across time points. Mean correlation was computed for each of the four regulatory mechanisms (Enhancer mono-pattern: 0.9, Silencer mono-pattern: 0.86, Enhancer poly-pattern: 0.68, and Silencer poly-pattern: 0.57). **Right panel:** Cross-timepoints MSE computed between the true and predicted values for each gene across time points. Mean MSE was computed for each of the four regulatory mechanisms (Enhancer mono-pattern: 0.87, Silencer mono-pattern: 0.39, Enhancer poly-pattern: 1.24, Silencer poly-pattern: 1.02).

Ref 2.6 - Minor points

Reviewer's comment	Line 44: "velocity of all genes" → rate of expression change of all genes Line 123: define "early" and "late" Line 131: k_i is a rate constant, not a rate. It should also not be interpreted as such.
Author's response	We thank the reviewer for suggesting these changes, and we have now addressed them with textual edits.

References for responses to Reviewer 2

Love, M. I., Huber, W. & Anders, S. Moderated estimation of fold change and dispersion for RNA-seq data with DESeq2. *Genome biology* **15** (2014).

Chang, C. C. et al. Second-generation PLINK: rising to the challenge of larger and richer datasets. *GigaScience* **4** (2015).

La Manno, G. et al. RNA velocity of single cells. *Nature* **560**, 494–498 (2018).

Bergen, V., Lange, M., Peidli, S., Wolf, F. A. & Theis, F. J. Generalizing RNA velocity to transient cell states through dynamical modeling. *Nat Biotechnol* **38**, 1408–1414 (2020).

Chen, Z., King, W. C., Hwang, A., Gerstein, M. & Zhang, J. DeepVelo: Single-cell transcriptomic deep velocity field learning with neural ordinary differential equations. *Sci Adv* **8** (2022).

Li, C., Virgilio, M. C., Collins, K. L. & Welch, J. D. Multi-omic single-cell velocity models epigenometranscriptome interactions and improves cell fate prediction. *Nat Biotechnol* **41**, 387–398 (2023).

Choi, J. et al. Evidence for additive and synergistic action of Mammalian enhancers during cell fate determination. *eLife* **10** (2021).

Zhou, W. et al. Genome-wide prediction of DNase I hypersensitivity using gene expression. *Nat Commun* **8** (2017).

Reviewer 3

Ref 3.0 - Reviewer's summary

Reviewer's comment	Borsari and co-workers present a mathematical framework based on ordinary differential equations (ODEs) and machine/deep learning to analysis time-stipulated epigenetic and gene expression datasets. Although the topic is timely and such framework would be very useful to the community for investigating abnormal development, diseased tissue or perturbed conditions (e.g. gene KO, drug exposure), the proposed modeling framework suffers from limitations and questionable choices and the documentation of its assessment using real data is incomplete, making it difficult to draw any conclusions on the validity of the approach. Here are my detailed comments.
Author's response	We agree with the reviewer that the topic is timely and applicable and we thank the reviewer for drawing our attention to limitations and questionable choices that needed to be further improved. In the revised manuscript we extended our algorithm to capture more signals with biological meaning, including those at the single cell level, where a known limitation is the sparsity of the data. We further improved the model performance by using the actual values that are the direct output of the ODE fitting step, instead of the derivatives. We hope these substantial revisions address the challenges associated with the complexity and inherent noise of biological data.

Ref 3.1 - Linear interpolation

Reviewer's comment	About the ODE modeling of cCREs: The first step of data interpolation before fitting the ODE model is very unusual and should not be pursued as it inflates the model-to-data error, dilutes the importance of real data points among synthetic ones, and consequently messes up the MSE computations. In other words, such preliminary step transforms the input data in a way that worsens the model fit to true data. I strongly suggest removing it as ODE model calibration can be performed directly on the datapoints, as widely done by the community.
Author's response	We thank the reviewer for this comment and we agree that this procedure would dilute the importance of real data points among synthetic ones. As requested, in our revised manuscript we have removed this step from our pipeline.

Ref 3.2 - Relative vs. actual values of gene/cCRE signals

Reviewer's comment	The reason why cCRE data is normalized is unclear. The actual absolute value of the signal may contain information which is lost through this normalization step. And this information can be integrated in the framework by estimating y_0 actual value, (possibly after a rescaling to ensure that all data points are positive). This will bring additional value to the framework which could offer to analyze not only relative cCREs values as it is the case now, but also absolute time-profiles, thus allowing for comparison between cCREs. Importantly, this may allow to use the ODE logistic model as an input in the machine/deep learning models (see comment below).
Author's response	We appreciate the reviewer's feedback. This is an excellent point, and in the revised version of the manuscript we have made significant adjustments to chronode to address these concerns. First, as requested, we have added both a Methods section and a section in the Supplementary Note explaining the necessity of data normalization. Briefly, this step avoids numerical instability when looking for an ODE solution for biological signals like gene/cCRE signals, which are positive values in the range 0-b. In fact, as we explain in the excerpt below, finding a fitting solution (k and b parameters) in the original range of values might be mathematically challenging in cases where $y_{min} \gg 0$ because the space of solutions is very limited and this raises numerical instability. Second, we have incorporated a post-fitting step in the revised manuscript, as per the reviewer's suggestion. This step shifts the data back to its original range, thereby preserving its biological interpretation while maintaining the advantages of normalization during the fitting process.
Excerpt from manuscript	Excerpt from Methods: Data normalization. When modeling biological signals that naturally lie in the positive range, the logistic function is expected to map input values to the range $[0,b]$, with symmetric acceleration and deceleration phases around $\frac{b}{2}$ (Fig. 1). However, direct fitting to the standard logistic function can become challenging when the vector of biological signals \vec{y} spans a range that deviates from this natural form. Specifically, when \vec{y} lies in a range where $y_{min} > 0$ (e.g., a range $[2, 5]$), the assumption of a lower bound at 0 is violated. This can lead to suboptimal fitting results, numerical instability during parameter optimization, and reduced flexibility in capturing fine-grained patterns (for more details, see Supplementary Note, "Note on data normalization"). To address this issue, we propose rescaling the data to ranges that align better with the theoretical framework of the logistic function. [...] Data rescaling to original range. To preserve the biological interpretation of y signals, we introduced a rescaling step following ODE fitting. In this step, the selected fitted

curve is rescaled back to its original range, ensuring that the data's original scale is retained to enable comparisons across genes and cCREs. [...]"

Excerpt from Supplementary Note (Note on data normalization and Proposition 5):

Note on data normalization

Consider a vector of the original data points \vec{y} containing p measurements over time t . The minimum and maximum signals in \vec{y} are denoted as y_{\min} and y_{\max} , respectively. We aim to fit a logistic curve on \vec{y} following:

$$y = \frac{bCe^{kt}}{b + Ce^{kt}}$$

where $C = \frac{y_0}{(1 - \frac{y_0}{b})e^{kt_0}}$.

In the context of biological data, the solution of a logistic curve fitting \vec{y} is constrained to the range $[0, b]$. When finding the solution, the parameter b can be adjusted to match the upper bound of \vec{y} , but the logistic function's lower bound is fixed at 0. Consequently, the range of \vec{y} directly influences the feasible solutions for the parameters k and b in the ODE. This restriction forces the optimization algorithm to operate in a limited and less informative region of the parameter space, potentially causing numerical instability.

For instance, consider a dataset \vec{y} with a range $[y_{\min}, y_{\max}]$ that spans only a limited portion of the logistic curve and lies entirely within the convex region of the curve, where $y_{\min} \gg 0$. As a result, the second derivative satisfies $y'' > 0$ throughout the interval $[y_{\min}, y_{\max}]$, such that:

$$y'' = k - \frac{2ky}{b} > 0$$

We observe that:

$$y'' > 0 \Rightarrow y < \frac{b}{2}, \forall y \in [y_{\min}, y_{\max}]$$

Thus, the space where to find parameter b is very limited, since b must satisfy:

$$b > 2y_{\max} \gg 0$$

This large b reduces the flexibility of the logistic curve, making it less capable of capturing finer local patterns in the data, ultimately leading to suboptimal fitting results.

To address numerical instability and improve fitting accuracy, the data points in \vec{y} are translated to fit within a normalized range near zero. The normalization process reduces the dynamic range of the data, lowering the magnitude of derivatives and alleviating ill-conditioning in the optimization problem. Once the logistic curve is fitted to the scaled data, the original scale and position can be restored using the following rescaling function:

$$\vec{y}_{\text{rescaled}} = \frac{(\vec{y}_{\text{fitted}} - R_{\min})(y_{\max} - y_{\min})}{R_{\max} - R_{\min}} + y_{\min}$$

where R_{\min} and R_{\max} represent the lower and upper bounds of the range to which the data was normalized ($[R_{\min}, R_{\max}]$) and \vec{y}_{fitted} is the fitted curve under the normalized range, a function of t . y_{\max} and y_{\min} represent the maximum and minimum values for the original data points.

Proposition 5. *The key properties of the curve (inflection point and kinetics class) will be preserved under the rescaling function.*

Proof. The inflection point is the point where the second derivative is zero. We have its first derivative with respect to t is:

$$\frac{d\vec{y}_{\text{rescaled}}}{dt} = \frac{y_{\max} - y_{\min}}{R_{\max} - R_{\min}} \frac{d\vec{y}_{\text{fitted}}}{dt}$$

The second derivative of the re-scaled function is:

$$\frac{d^2\vec{y}_{\text{rescaled}}}{dt^2} = \frac{y_{\max} - y_{\min}}{R_{\max} - R_{\min}} \frac{d^2\vec{y}_{\text{fitted}}}{dt^2}$$

As $\frac{y_{\max} - y_{\min}}{R_{\max} - R_{\min}}$ is just a constant, we have:

$$\frac{d^2\vec{y}_{\text{rescaled}}}{dt^2} = 0 \text{ if and only if } \frac{d^2\vec{y}_{\text{fitted}}}{dt^2} = 0$$

□

Based on this demonstration, we can rescale the fitted curve back to its original range without affecting the switching point; therefore, the key properties of the fitted curve remain under data normalization and rescaling.

Ref 3.3.a - Filtering ODE-fitted signals based on MSE

Reviewer's
comment

The ODE fitting assessment is not properly reported which makes it impossible to assess the validity of the model, and of the choice of the logistic function. The authors only report MSE values which are relative to

	a given problem and do not provide the information of what is a good or a bad fit. We need to see the MSE distribution over all cCRE and examples of what is acceptable/unacceptable. The authors states that they excluded the 20% of elements with the highest reported MSE but this threshold is not justified.
Author's response	We thank the reviewer for this comment and agree that our previous approach to identifying good fits was somewhat arbitrary. As requested, we have adopted a more rigorous method in the revised manuscript. We now assess the distribution of MSE values using Gaussian Mixture Models (GMMs), which are commonly used in anomaly detection tasks (Liu et al 2019, J. Phys). This approach assumes that the MSE distribution consists of two groups: "acceptable fits" (low MSE) and "unacceptable fits" (high MSE). The GMM fits two Gaussian distributions to the data, assigning each gene or cCRE a probability of belonging to either group. The algorithm then determines a cutoff threshold based on these probabilities to separate the two clusters, and genes with MSE values exceeding this threshold are considered poor fits and filtered out. Applying this methodology, we found that, on average, 13% of dynamic genes and 10% of dynamic cCREs were filtered out as poor fits across the three brain regions, indicating that their time-series patterns could not be adequately captured by the logistic ODE. These excluded genes often exhibited abrupt changes (peak-like pattern) or noisy signals that the logistic fits struggled to represent. The insights from this filtering process, combined with the reviewer's constructive feedback, have led us to refine our fitting strategy to better handle these complex time-series profiles (see next comment). We have added an explanation of this approach in the Methods and included the distributions of gene expression MSE in new Supplementary Figure 2D-E together with examples of acceptable and unacceptable fits (see excerpts below).
Excerpt from manuscript	Excerpt from Methods section: "Gaussian Mixed Modelling (GMM) has been previously used for anomaly detection tasks⁶⁵. Based on this, here we used a GMM-based approach to classify genes and cCREs as "acceptable" or "unacceptable" fits based on the distribution of their Mean Squared Errors (MSEs). Accordingly, we model the MSE distribution by two Gaussian components: one representing acceptable fits with low MSE values, and another representing unacceptable fits with high MSE values (Supplementary Fig. 1), assigning each data point (gene or cCRE) a probability of belonging to each group. A cutoff threshold is then determined to distinguish between the two clusters. Data points with MSE values exceeding this threshold were classified as unacceptable fits. Specifically, for a given dataset containing either genes' or cCREs' ODE-fitted genomic signals rescaled to the original range, we utilized the function <code>normalmixEM()</code> from the R package <code>mixtools</code>⁶⁶ (setting parameter <code>k=2</code>) to fit the GMM, and the function <code>plot_cut_point()</code> from the R package <code>plotGMM</code> to calculate the cutoff threshold for acceptable MSE values. To showcase this approach in the case of monotonic fits for bulk

RNA-seq data, in **Supplementary Fig. 2D-E** we reported the GMM-partitioned MSE distributions for the three brain regions as well as examples of genes with acceptable and unacceptable fits.”

Excerpt from new **Supplementary Figure 2D-E**:

Supplementary Figure 2: Monotonic and piecewise fitting of gene expression kinetics during mouse brain development. **D:** Distribution of Mean Squared Error (MSE) of monotonically fitted genes across the three brain regions. The distribution is partitioned into acceptable (red) and unacceptable (cyan) fits following Gaussian Mixed Model (GMM) analysis. The GMM cutoff is indicated by a dashed vertical line. **E:** Examples of acceptable and unacceptable fits with the corresponding MSE values. The x axis represents PC Days, and the y axis shows gene expression values expressed in Transcripts Per Million (TPMs, \log_2 -transformed). The black dots correspond to the data points prior to monotonic fitting, and the grey line corresponds to the monotonically fitted expression profile.

Ref 3.3.b - Non-logistic curves

Reviewer's
comment

The subsequent question is about the reason of the bad fits and whether they support to introduce a more complex model with additional degrees of freedom in your model. Indeed, rejected cCRE time-profiles from data analysis because they look more complex than a logistic curve is not a biologically-sound argument, and the framework must handle such cases. In other words, the modeling needs to adapt to the data, and not the other way round.

Author's response

We agree with this comment and thank the reviewer for this insightful suggestion. We acknowledge that our previous filtering strategy restricted our ability to capture biologically meaningful patterns beyond those modeled by the logistic curve. In response, we have revised our approach to allow for greater flexibility in detecting more complex, biologically relevant patterns, such as peak-like signals. Our updated pipeline now enables non-monotonic signals to be fitted by multiple logistic functions. Specifically, we demonstrate this by fitting two logistic functions to capture peak-like patterns. Additionally, we have expanded our classification to include a new group of patterns (e.g., piecewise fits) and present these updated results in new Supplementary Figure 3C-E (see excerpt below).

Excerpt from manuscript

Excerpt from main manuscript (Results):

"We note that this approach is well-suited for modeling genomic signals y that exhibit monotonic increases or decreases. While these monotonic fits describe most of the observed transcriptional programs, a small fraction of genes may display peak-like expression profiles over time, such as those involved in circadian clocks, cell cycle regulation, or stress responses³⁰⁻³². Thus, we further extended our methodology by introducing a piecewise fitting approach that can accommodate such profiles by combining two logistic curves (or two portions of them). Although alternative functions (e.g., Gaussian curves) could be employed for these cases, our choice to use logistic functions maintains consistency within the framework and allows for comparable kinetic parameters across both monotonic and piecewise fits. Both approaches are implemented as standalone pipelines, providing a scalable solution for genome-wide applications at both bulk and sc levels."

Excerpt from new **Supplementary Figure 3C-E**:

Supplementary Figure 3: Properties of Q1-Q4 genes. **C:** Distribution of the kinetic parameters k (magnitude expressed in absolute value, y axis) and b (x axis) across all genes with piecewise fits in the three brain regions. For genes with piecewise sigmoid fits, two k values are computed (k_{left} and k_{right} , one for each sigmoid segment) and here we display their average absolute k value (k_{avg}). The plot subdivision in four Q1-Q4 quadrants is the same shown in Figure 2A. **D-E:** Lineplot showing, for genes modelled by piecewise fits in Q1 through Q4, the average expression (y axis) over time (x axis; PC = Post-Conception). Expression profiles with concave peaks are characterized by $k_{left} > 0$ (**D**), while those with convex peaks have $k_{left} < 0$ (**E**). Note that the average

	gene expression levels (expressed in log ₂ TPMs) were rescaled to the range 0-100% to allow for comparison across Q1-Q4 groups.
--	--

Ref 3.4 - Using cCRE derivatives as input for the models

Reviewer's comment	About the machine/deep learning models: It is unclear why the authors chose to use the derivatives of cCRE (dCi/dt) as inputs instead of directly using the amount of CRE (Ci). Indeed, corresponding ODE models would have used linear (e.g. law of mass action) or non-linear (e.g. Hill functions) terms involving Ci, and not dCi/dt. This choice adds the complexity of estimating dCi/dt which is not directly documented in the data and needs to be approximated in an additional step that is associated with additional error (see also next comment).
Author's response	We thank the reviewer for this comment and agree that estimating dCi/dt added unnecessary complexity to the model. As requested, in our revised approach, we now use the actual chromatin values (Ci) as input to the model, rather than the derivatives. Similarly, we now predict gene expression values as the output, instead of the expression derivatives, simplifying the prediction task and reducing associated errors.
Excerpt from manuscript	Excerpt from main manuscript (Results): “At each time point t ($t \in \{1, 2, \dots, 8\}$) we define $\vec{c}_{i,t}$ as the vector of m cCRE signals: $\vec{c}_{i,t} = [c_{i,t,1}, c_{i,t,2}, \dots, c_{i,t,m}]$ [...] to predict \vec{g}_i i.e., the gene expression profile over time: $\vec{g}_i = [g_{i,0}, g_{i,1}, \dots, g_{i,t}]$ [...]”

Ref 3.5 - Linear interpolation for estimating the derivatives

Reviewer's comment	For estimating the derivatives of cCRE amount (dCi/dt), the authors have used linear interpolation, which is a bad choice, as it propagates the experimental error (see my comment above). Considering that the first part of this study is dedicated to the design of an ODE model that precisely provides the derivative of Ci, it is surprising that the authors did not use it as input of their deep learning regressors. Indeed, contrarily to linear interpolation, such ODE-based model tends to generate smoother profiles, that may even correct for part of the experimental error when present in only one data point of time-profile, for instance.
--

Author's response	We thank the reviewer for this insightful suggestion and agree that linear interpolation is suboptimal in an ODE-based model, which typically produces smoother profiles. As requested, we have removed the linear interpolation step from our pipeline.
-------------------	--

Ref 3.6 - Performance of the predictive models

Reviewer's comment	Performances of the proposed RF or NN models are low as indicated by low correlation between model and data. More importantly, the models create peaks or troughs in the time-profiles which were not present in the data, thus impairing the subsequent biological interpretation of the data (Fig5c, sup. Fig. 7). As they are now, such unreliable algorithms may not be used by the community.
Author's response	We thank the reviewer for this comment. In response, we have made substantial revisions to our model to incorporate the reviewer's suggestions. Specifically, we have: 1) used actual cCRE chromatin accessibility values and gene expression as the input and output of the model, respectively; 2) expanded the model to integrate signals from multiple cCREs for predicting gene expression; and 3) incorporated a bi-directional Recurrent Neural Network (biRNN) to better capture time-dependent dynamics. Our updated approach now predicts gene expression over time across four distinct regulatory mechanisms (enhancer mono-p. genes, silencer mono-p. genes, enhancer poly-p. genes, and silencer poly-p. genes), offering an advancement beyond current state-of-the-art methods that focus mostly on mono-pattern activators (Mitra et al., 2024 Nat Genet). Regarding the model's performance, we have now clarified this aspect in the revised text, specifying that the goodness of prediction is relative to the state of the art. We evaluated this in two ways: first, we computed the cross-gene correlation between true and predicted values across all genes, achieving values ranging from 0.87 to 0.97 (Figure 5B), which are higher than those previously reported in the literature. For example, Zhou et al. (2017, Nat Commun) reported values between 0.65 and 0.82 for a similar prediction task (i.e., predicting chromatin accessibility from gene expression across cell types). Second, we computed the cross-timepoint correlation between true and predicted values for each gene, which resulted in mean correlation values ranging from 0.57-0.90, again higher than the ones reported by Zhou et al. (in the 0.35–0.39 range).
Excerpt from manuscript	Excerpt from main manuscript (Results): "To evaluate the performance of the model, we first computed cross-gene correlations between true and predicted expression values. Our predictions on the test set showed a strong positive correlation (>0.85) in all the four types of regulatory mechanisms (Fig.

5B), higher compared to previous reports of chromatin-gene expression predictions⁴⁴. We also evaluated the model performance by computing the distribution of Mean Squared Error (MSE) and cross-timepoint correlation between true and predicted gene expression values (**Supplementary Fig. 7C**). In particular, we achieved mean cross-timepoint correlations ranging between 0.57 and 0.90, higher than the equivalent cross-cell type correlations reported previously⁴⁴. [...]"

Excerpt from new **Figure 5B**:

Figure 5: Predicting time-series gene expression from chromatin signals of associated cCREs. **B:** Density plot showing cross-gene expression correlation for each of the four regulatory mechanisms. Each scatterplot shows true (x axis) versus predicted (y axis) expression values (\log_2 -transformed TPMs) across all genes in the test set.

Excerpt from new **Supplementary Figure 7C**:

Supplementary Figure 7: biRNN Predictions across four gene regulatory mechanisms. **C. Left panel:** Cross-timepoints correlation computed between the true and predicted values for each gene across time points. Mean correlation was computed for each of the four regulatory mechanisms (Enhancer mono-pattern: 0.9, Silencer mono-pattern: 0.86, Enhancer poly-pattern: 0.68, and Silencer poly-pattern: 0.57). **Right panel:** Cross-timepoints MSE computed between the true and predicted values for each gene across time points. Mean MSE was computed for each of the four regulatory mechanisms (Enhancer mono-pattern: 0.87, Silencer mono-pattern: 0.39, Enhancer poly-pattern: 1.24, Silencer poly-pattern: 1.02).

Ref 3.7 - Implementing an architecture suitable for time-series predictions

Reviewer's comment	The proposed machine/deep learning models are rather simple whereas more complex model structure should be employed here to properly predict gene time profiles. Indeed, the authors claims that the specificity of the framework lays in the analysis of time-stipulated data whereas the RF or NN models do not account for this time structure of the data, which may explain the large discrepancies observed between the true and predicted gene profiles. Authors should consider using NN framework like neural ODEs that are likely to give better results for this specific problem.
Author's response	We thank the reviewer for this suggestion. We agree with the reviewer that RF models are not suitable for time-series analysis and we have removed this part of the analysis from the revised version of the manuscript. To further address this comment, we have now introduced a more complex NN architecture that is specifically suited to handle time-series data and can also accommodate multiple cCRE signals per gene (and potentially more epigenetic features besides chromatin accessibility) in the form of a tensor. Additionally, we are now using a bidirectional RNN cell to better capture the complex interplay between chromatin and gene expression. This approach accounts for multiple possible scenarios: changes in chromatin accessibility at cCREs upstream of the promoter can influence gene expression, for example, by enabling transcription factor binding or facilitating Pol II release. Conversely, transcriptional activity—through Pol II elongation—can also induce changes in chromatin accessibility at both upstream and downstream cCREs. The biRNN is particularly well-suited for this task, as it leverages both past and future time points to make predictions based on the full temporal context. Overall, we have observed increased predictive performance compared to the previous version of the NN model.

Excerpt from manuscript

Excerpt from main manuscript (Results):

“Building on these findings and utilizing our many-to-one linking schema—where multiple cCREs are linked to a given gene over time—we designed a model architecture (Fig. 5A) capturing four types of gene regulatory mechanisms [...].

Our modeling approach predicts gene expression at each time point through regression, using a bidirectional RNN-based architecture which is well suited for time-series data. [...]

We use a biRNN to model gene expression $g_{i,t}$ at a particular time point t , as a function of the cCRE chromatin accessibility signal vector $\vec{c}_{i,t}$. In fact, capturing the interplay between chromatin and gene expression is not a trivial task, since multiple scenarios are possible. For instance, changes in chromatin accessibility at cCREs upstream of the promoter can potentially influence gene expression, for example, by enabling transcription factor binding or facilitating Pol II release. Conversely, transcriptional activity—through Pol II elongation—can also induce changes in chromatin accessibility at both upstream and downstream cCREs. The biRNN allows us to cover this wide range of scenarios by leveraging chromatin signals at both past and future time points to make predictions of gene expression. [...]

We note that the usage of the biRNN is especially effective for predicting the expression of poly-pattern genes, which are particularly challenging to model due to the mixed contributions of enhancer and silencer cCRE signals (Supplementary Fig. 7D). The biRNN leverages the more comprehensive learning context provided by both past and future time points to address this complexity. Additionally, incorporating a dense layer enhances the model's capacity to learn complex kinetic relationships between cCREs and their associated genes.”

Excerpt from new Figure 5A:

Figure 5

	Figure 5: Predicting time-series gene expression from chromatin signals of associated cCREs. A: Architecture of the biRNN-based model. At each time point, the input for a specific multicRE-gene pair i is a vector \vec{c} of dimension $1 \times m$, where m represents the number of cCRE signals associated with the gene. The model generates an output vector, representing the gene expression levels \vec{g} across all time points. Excerpt from new Supplementary Figure 7D: D  Supplementary Figure 7: biRNN predictions across four gene regulatory mechanisms. D: Distributions of cross-timepoint Pearson's correlations between true and predicted gene expression values using a linear (orange) or non-linear (green) model. The baseline linear model corresponds to a neural network (NN) without bidirectional RNN (biRNN) and ReLU activation. Statistically significant differences between distributions were assessed using the Wilcoxon paired test: ns = not significant (p value > 0.05), * = p value < 0.05, ** = p value < 0.01. For poly-pattern genes, the linear NN shows significantly lower correlations than the non-linear NN (enhancer poly-pattern: p value = 0.0043; silencer poly-pattern: p value = 0.0285). No significant differences were observed for mono-pattern genes (enhancer mono-pattern: p value = 0.736; silencer mono-pattern: p value = 0.2227).
--	---

Ref 3.8 - Using multiple cCREs to predict a gene's expression

Reviewer's comment	Assuming that the expression of a gene may be predicted by a single cCRE is a questionable choice. It should probably be relaxed to increase model performance.
Author's response	We thank the reviewer for this comment and agree that predicting gene expression by a single cCRE is a questionable choice. As requested, we have now expanded the prediction of gene expression using multiple cCREs per gene, providing stronger biological reasoning along with higher model performance (see comments 3.6 and 3.7).

Ref 3.9 - RF hyperparameters

Reviewer's comment	RF models are not properly trained as hyperparameters (e.g. number of trees) should have been determined by cross-validation.
Author's response	We thank the reviewer for this comment. In the previous version of the manuscript, the RF hyperparameters were determined using cross-validation. That said, we have decided to remove the RF model from our revised version as we think the RF is less interpretable for time series data - as suggested by the reviewer - and especially in this case

	where we want to model the joint input effect of multiple cCREs.
--	--

References for responses to Reviewer 3

Liu, J. et al. Anomaly detection for time series using temporal convolutional networks and Gaussian mixture model. *Journal of Physics: Conference Series* **1187**, 042111 (2019).

Mitra, S. et al. Single-cell multi-ome regression models identify functional and disease-associated enhancers and enable chromatin potential analysis. *Nat Genet* **56**, 627–636 (2024).

Zhou, W. et al. Genome-wide prediction of DNase I hypersensitivity using gene expression. *Nat Commun* **8** (2017).

Reviewer #3

The authors have responded to most of my concerns. Please find below some remaining questions.

Major points

Ref 3.1

Reviewer's comment	Related to Ref3.2: The current model only partly recapitulates the data as the absolute values of the gene levels are not present in the k and b parameters. The information is in the R_{min} and R_{max} scaling factors and the minimum the authors could do is to present statistics of these factors. The fact that the authors need to normalize the data to fit them suggests that the model is not flexible enough and would benefit from the addition of an extra parameter allowing for the modification of the value of the lower asymptote of the curve.
Author's response	We thank the reviewer for this comment. As the reviewer mentions, fully fitting the data within its real (i.e., observed) range would require additional flexibility, specifically parameters that allow adjustment of the lower asymptote of the curve. In the revised manuscript, we now show that we can fully recapitulate the data using a modified version of the ODE that accommodates this flexibility: $\frac{dz}{dt} = k(z - a)\left(1 - \frac{z - a}{b - a}\right)$ However, fitting this general form directly, although more flexible, involves more parameters and thus can lead to instability, particularly when the number of time points is limited. In fact, with only eight data points, estimating three parameters becomes unstable: that is, the model with three parameters exhibits a higher susceptibility to large convergence errors during parameter fitting, likely due to the increased complexity and flexibility of the parameter space, as explained by Motulsky & Christopoulos (2004, Chapter A "Fitting data with non-linear regression")¹. Instead, to enhance stability, we fit the translated (or simplified) form of the ODE previously introduced in the manuscript: $\frac{dy^*}{dt} = k^*y^*\left(1 - \frac{y^*}{b^*}\right)$ This form of the ODE can be used to reduce the model from three to two parameters fixing the lower asymptote to zero (i.e., $a^* = 0$), improving

	numerical stability while essentially preserving the kinetic features. This allows us to determine the rate constant k with far more precision¹. Since the parameter a (which corresponds to the lower asymptote in the first ODE) cannot always be inferred from the data, we perform the fitting on normalized data which better meets the assumption of the simplified ODE of having a lower asymptote equal to zero (see Supplementary Note, Propositions 1 and 6, excerpted below). To address the reviewer's concern regarding the interpretation of parameters k^* and b^*, we now make clear in our nomenclature the distinction between k & b (in the observed space), and k^* & b^* in the simplified (e.g., normalized) space. In Propositions 7-8 (Supplementary Note, excerpted below), we now explicitly show that both translation and normalization of the data preserve the value of k as it would appear in the real (observed) data range (see also new Supplementary Fig. 2, excerpted below). Additionally, we now explicitly demonstrate that both a and b in the real range can be recovered using an inverse transformation (see Supplementary Fig. 2 and Methods section "Restoring the fitted curve to the original range of the data", excerpted below). In summary, we demonstrate that the parameters k^* and b^* can be interpreted in the context of the original data. We also note that the distributions of b shown in Figures 2A and 3A (as well as previous Supplementary Fig. 3C, now Supplementary Fig. 4C) already correspond to the b values in the original range. Finally, as requested, we now include summary statistics of the scaling factors in the revised manuscript (see new Supplementary Fig. 3F and Supplementary Table 4).
Excerpt from manuscript	Excerpt from Supplementary Note (Propositions 1, 6, 7, and 8):

Proposition 1. The time-series signal of a given gene or cCRE is modeled using the following generalized logistic ordinary differential equation (ODE):

$$\frac{dz}{dt} = k(z - a)\left(1 - \frac{z - a}{b - a}\right)$$

Here, $z(t)$ denotes the gene or cCRE signal at time t , constrained within the interval $[a, b]$. The parameters a and b correspond to the lower and upper asymptotes of the logistic curve, respectively. k is a rate parameter that determines the steepness of the curve. Let $z_{start} = z(t_{start})$ denote the observed signal at the first time point monitored by the time-course experiment.

The analytical solution of this ODE is (see **Proposition 5**):

$$z(t) = \frac{(b - a)Ce^{kt}}{1 + Ce^{kt}} + a.$$

However, fitting this general form directly, although more flexible, can lead to potential numerical instability. In fact, with only eight time points (like in the case of the bulk data used in this paper), estimating the three parameters a , b , and k becomes unstable. In other words, the model with three parameters exhibits a higher susceptibility to large convergence errors during parameter fitting, likely due to the increased complexity and flexibility of the parameter space (see also Motulsky & Christopoulos, 2004, Chapter A "Fitting data with non-linear regression")¹.

To mitigate this issue, we consider the simplified form of the logistic ODE:

$$\frac{dy^*}{dt} = k^*y^*\left(1 - \frac{y^*}{b^*}\right).$$

Note that this equation is a translated version of the generalized form via shifting relative to a (i.e., $z - a = y$), which simplifies the algebraic expression. This allows to reduce the model from three to two parameters, fixing the lower asymptote to zero (i.e., $a = 0$) and improving numerical stability while essentially preserving the kinetic features. In this way, we can determine the rate constant k with far more precision¹.

However, knowing y requires prior knowledge of a . To circumvent this dependency, we instead fit the simplified ODE on a normalized version of y (e.g., y^*), which is equivalent to the normalized version of z (see **Proposition 6** and **Supplementary Fig. 2**). In fact, the normalized data y^* better meets the assumption of the ODE of having a lower asymptote equal to zero (i.e., $a^* = 0$).

[...]

Proposition 6. Normalizing y to the range $[R_{min}, R_{max}]$ is equivalent to normalizing z to the range $[R_{min}, R_{max}]$.

Proof. We define y^* in the normalized range as:

$$y^* = \frac{(y - y_{min})(R_{max} - R_{min})}{y_{max} - y_{min}} + R_{min}$$

with the following relationships:

$$\begin{aligned} y &= z - a; \\ y_{min} &= z_{min} - a; \\ y_{max} &= z_{max} - a. \end{aligned}$$

Substituting into the expression for y^* :

$$\begin{aligned} y^* &= \frac{((z - a) - (z_{min} - a))(R_{max} - R_{min})}{(z_{max} - a) - (z_{min} - a)} + R_{min} \\ &= \frac{(z - a - z_{min} + a)(R_{max} - R_{min})}{z_{max} - a - z_{min} + a} + R_{min} \\ &= \frac{(z - z_{min})(R_{max} - R_{min})}{z_{max} - z_{min}} + R_{min} \end{aligned}$$

In summary, the normalization cancels out the shift by a , so y^* depends only on z . Thus, normalizing y gives the same result as normalizing z (see **Supplementary Fig. 2**). □

Proposition 7. *The kinetic parameter k is preserved under translation of the logistic ODE.*

We consider the following logistic ODE:

$$\frac{dy}{dt} = ky \left(1 - \frac{y}{b_y}\right),$$

where $y > 0$ and $b_y > y$.

Given an initial condition (t_{start}, y_{start}) , where y_{start} is the first point in the range of an experimental time-course data sample $y \in [0, b_y]$, we aim to show that this equation is equivalent to a translated version of itself.

Proof. To introduce a translation, we define:

$$z - a = y,$$

where a is a constant shift.

Differentiating both sides with respect to t :

$$\frac{dz}{dt} = \frac{dy}{dt}.$$

Substituting into the original logistic equation:

$$\frac{dz}{dt} = ky \left(1 - \frac{y}{b_y}\right).$$

Now rewrite in terms of z using $y = z - a$, and $b_y = b - a$:

$$\frac{dz}{dt} = k(z - a) \left(1 - \frac{z - a}{b - a}\right).$$

where $z \in [a, b]$.

In summary, shifting by a , i.e., transforming the domain from the observed range $[a, b]$ to the translated range $[0, b_y]$, preserves the logistic form and keeps the parameter k unchanged (see **Supplementary Fig. 2**). □

Proposition 8. *The kinetic parameter k is preserved under min-max normalization.*

For a given gene or cCRE, let $z(t)$ be a sigmoid function modeling its time-series profile, bounded in the observed data range $[a, b]$ over the time interval $t \in [t_{start}, t_{end}]$:

$$a \leq z(t) \leq b, \quad \forall t \in [t_{start}, t_{end}]$$

We define k as the kinetic parameter in the observed range of the data $[a, b]$, and k^* as the corresponding parameter in the normalized range $[0, b^*]$.

We aim to show that the parameter k is invariant under a linear transformation, specifically under min-max normalization, i.e., $k = k^*$.

Proof. We start with the generalized form of the logistic ODE in the observed range of the data:

$$\frac{dz}{dt} = k(z - a) \left(1 - \frac{z - a}{b - a}\right)$$

Assuming the normalized data in the range $[0, 1]$ represents a portion of the full sigmoid curve, let $z^* \in [0, b^*]$ be the normalized version of z obtained with min-max normalization, as follows:

$$z^* = \frac{(z - a)(b^* - 0)}{(b - a)}$$

or equivalently:

$$z = \frac{z^*(b - a)}{b^*} + a.$$

Differentiate z with respect to time t :

$$\frac{dz}{dt} = \frac{dz^*}{dt} \frac{b - a}{b^*}$$

Substitute this and the expression for z into the original ODE:

$$\frac{dz^*}{dt} \frac{(b - a)}{b^*} = k \frac{(z - a)(b - a)}{b^*} \left(1 - \frac{z - a}{b - a}\right)$$

Simplify and obtain the simplified form of the ODE that applies to the normalized range of the data:

$$\frac{dz^*}{dt} = kz^* \left(1 - \frac{z^*}{b^*}\right)$$

In summary, the ODE retains the same form in the normalized range, with the same kinetic parameter k (that is, $k^* = k$). Therefore, k is invariant under min-max normalization (see **Supplementary Fig. 2**). □

Excerpt from main text (Results section):

“Given the cooperative and saturating nature of genomic signals (**Fig. 1**), we propose modeling transcriptomic and epigenomic kinetics over time using the logistic function, previously applied in fields like bacterial growth²⁸⁻²⁹. In this context, the signal of a genomic locus (e.g., gene expression, chromatin accessibility, or histone modification) is

defined as a time-dependent positive variable z , constrained within the interval $[a, b]$. The rate of change of z over time t can be studied using the following formula which corresponds to the generalized form of the logistic ODE (see **Supplementary Note**, Propositions 1 & 5, and **Supplementary Fig. 2**):

$$\frac{dz}{dt} = k(z - a)\left(1 - \frac{z - a}{b - a}\right) \quad (1)$$

where a and b correspond to the lower and upper asymptotes, respectively, of the logistic curve in the real range of the data.

However, fitting this general form directly can lead to instability, particularly when the number of experimental time points available is limited³⁰. To mitigate this and provide a simpler and easier-to-understand perspective, we fit the data using a “simplified” form of the ODE (after appropriate translation and normalization of the data; **Supplementary Note**, Propositions 6-9, and **Supplementary Fig. 2**):

$$\frac{dy^*}{dt} = k^* y^* \left(1 - \frac{y^*}{b^*}\right), \text{ with } y^*(t_{start}) = y_{start}^* \quad (2)$$

where t_{start} represents the initial time point of the experimental time course, and y_{start}^* represents the normalized gene expression level or chromatin signal of the genomic locus at t_{start} .”

Excerpt from Main Text (Discussion section):

“The fitting process uses a simplified form of the ODE with two parameters, instead of the generalized form of the ODE with three parameters, improving both interpretability and stability when working with a small number of data points.

[...]

To overcome the limitations of fitting ODEs with more than two parameters with a limited number of data points available, incorporating regularization may offer a promising direction to enhance both flexibility and stability.”

Excerpt from Methods

“Restoring the fitted curve to the original range of the data.

To preserve the biological interpretation of \vec{y}^* signals, we apply an inverse transformation of the fitted parameters a^* and b^* to obtain a and b , respectively, using the following formula (see **Supplementary Note**, Proposition 6, and **Supplementary Fig. 2**):

$$\begin{aligned} a &= \frac{(a^* - R_{min})(z_{max} - z_{min})}{R_{max} - R_{min}} + z_{min} \\ b &= \frac{(b^* - R_{min})(z_{max} - z_{min})}{R_{max} - R_{min}} + z_{min} \end{aligned} \quad (10)$$

where R_{min} and R_{max} represent the lower and upper bounds of the range to which the data was previously normalized ($[R_{min}, R_{max}]$). For example, $R_{min} = 10^{-5}$ and $R_{max} = 1$ for data normalized in the $[10^{-5}, 1]$ range, $R_{min} = 1$ and $R_{max} = 2$ for data normalized in the $[1, 2]$

range. Notably, the kinetic parameter k remains unchanged in the normalized and original ranges (i.e., $k^* = k$; see **Supplementary Note**, Propositions 7 & 8).

With the original z_{start} , and a and b determined, we use the analytical solution of the generalized logistic ODE (Equation 1) to recover the fitted curve in the original data range, denoted as $z(t)$ (see **Supplementary Note**, Proposition 5):

$$z(t) = \frac{(b-a)Ce^{kt}}{1+Ce^{kt}} + a. \quad (11)$$

Excerpt from new Supplementary Fig. 2:

Supplementary Figure 2. Transformation and fitting of a generalized logistic ODE function to model time-series functional genomics data. Top left: The general form of the ODE (Eq. 1) describing time-series gene expression or chromatin signal $z(t)$ is parametrized by three parameters (the rate constant k , and the lower and upper asymptotes a and b), and follows a logistic-like curve. Middle left: This general form is transformed into a “simplified” ODE with two parameters (k^* and b^* , assuming a lower asymptote $a^* = 0$). The time-series signal y^* is obtained upon min-max normalization of $z(t)$ (see Methods); y^*_{start} corresponds to the normalized value y^* measured at time point t_{start} (i.e., the first time point of the experimental time course). We numerically estimate the parameters k^* and b^* that best fit the data. In Propositions 7-8 (Supplementary Note), we show that both translation and normalization of the data from $z(t)$ to y^* preserve the value of k as it would appear in the real (observed) data range. On the other hand, a and b in the real range can be recovered using an inverse transformation (see Methods, “Restoring the fitted curve to the original range of the data”). Bottom right: The analytical solution $z(t)$ is directly derived using these parameters, allowing reconstruction of the smooth time-series functional genomics data trajectories, demonstrating that the parameters k^* and b^* from the simplified ODE can be interpreted in the context of the original data. Right: Examples of model fits (orange curves) to raw gene expression data (black dots). Note that these fits are the same shown in Fig. 2C.

Excerpt from new Supplementary Fig. 3F:

Ref 3.2

Reviewer's comment	Model fit: the authors need to provide the values of (t_0, y_0) they have used in the fit and their justification.
Author's response	We thank the reviewer for this comment. We realize the definition of t_0 and y_0 may have caused confusion, as they could be misinterpreted as referring to the lower asymptote of the curve and the time at which it is reached. To clarify, we have revised the manuscript to replace t_0 and y_0 with t_{start} and y^*_{start} , respectively. With this updated nomenclature, we aim to make it clearer that t_{start} (previously referred to as t_0) represents the first time point of the experimental time course. For instance, in our bulk gene expression and chromatin accessibility datasets, t_{start} corresponds to embryonic day 10.5, and this is encoded accordingly in the model fit. We also note that our data does not necessarily include measurements of the lower asymptote of the curve. The revised manuscript has been updated to make these distinctions and definitions clearer.
Excerpt from manuscript	Excerpt from Main Text (Results section): $\frac{dy^*}{dt} = k^* y^* \left(1 - \frac{y^*}{b^*} \right), \text{ with } y^*(t_{start}) = y^*_{start} \quad (2)$ where t_{start} represents the initial time point of the experimental time course, and y^*_{start} represents the normalized gene expression level or chromatin signal of the genomic locus at t_{start}.

	[...] We applied chronODE on the set of DE genes (defining, for a given gene, $t_{start} = 10.5$, and y^*_{start} = normalized expression level of the gene at time point 10.5; see Methods) [...].” Excerpt from Methods section: “Estimation of the kinetic parameters k and b. [...] In each of these four combinations, t_{start} is defined as the first time point monitored by the study (for instance, when applying chronODE on the bulk RNA-seq data from forebrain, $t_{start} = 10.5$), while y^*_{start} is defined as the expression level of a given gene in the forebrain at day 10.5 following the normalization step [...].”
--	---

Ref 3.3

Reviewer’s comment	1595: what are those y_0 values? Where do they come from since they are not estimated?
Author’s response	These values correspond to the original y_0 values from the data. Indeed, our approach involves solving an initial value problem. In the revised manuscript, we now refer to y_0 as y^*_{start} to clarify that it represents the normalized level of a given cCRE or gene at the starting point of the experimental time course (see previous comment).

Ref 3.4

Reviewer’s comment	Ref3.3b More details would be appreciated regarding the piecewise fit: how do the authors decide on the time cut-off between the logistic curves? Do they optimize it in the fit? Are the logistic curves fitted sequentially or simultaneously. To ensure the profile regularity, a constraint on the first derivative should be applied at the time cut-off. Regarding goodness of fit, do they use the same threshold for MSE as for monotonic profiles?
Author’s response	We appreciate the reviewer’s point that fitting peak-like patterns requires further details and clarifications. In brief, to preserve the usage of sigmoid functions also for fitting peak-like patterns and make it consistent with the rest of the manuscript, we wanted to perform the fitting just with sigmoid functions. Thus, we applied a piecewise fitting approach using two separate sigmoid functions, as follows. We selected the cut-off time point at the global minimum or maximum of the signal. The two logistic curves were then fitted sequentially. First, we fit the left segment. To ensure

	continuity of the function (0th-order continuity), we constrained the fitted peak point to be the starting point of the right segment's fit. This guarantees continuity of the fitted function itself, though not of its first derivative. Thus, our fitted peak-like curves have a discontinuity in the first derivative. However, we show that they are very close to what we would get with a standard B-spline fitting that ensures first-derivative continuity (see distribution of MSE values between ODE-based fit and the B-spline fit in new Supplementary Fig. 4F, excerpted below). In summary, our approach ensures continuity and has the advantage that it is kinetically interpretable because we use sigmoid functions parametrized by k and b parameters. We would like to emphasize that the main focus of our paper is to infer biological insights from the overall temporal kinetics, rather than from local effects. Accordingly, our method is designed primarily to analyze monotonic biological signals, which represent the majority of our data (~87%). In this context, peak-like fitting falls outside the main scope of this paper, and we did not further optimize this aspect. Instead, we present our current approach as a first step, and suggest that future studies could enhance the analysis of non-monotonic signals by detecting and modelling local maxima or minima. Regarding the question of goodness-of-fit for peak-like signals, we note that we do not use the same threshold as for monotonic profiles. Instead, we apply a threshold derived from GMM that was specifically computed based on the MSE values obtained from the piecewise fits. Finally, in the revised manuscript, we have expanded the discussion to include the advantages and limitations of our piecewise fitting approach, as well as potential directions for future improvements.
Excerpt from manuscript	Excerpt from Methods section: “For those cases with unacceptable monotonic fits, we next applied the logistic function in a piecewise manner, identifying the global maximum or minimum point of a given gene or cCRE signal. First, the left segment is fitted. To ensure zero-order continuity, the logistic function is adjusted so that the endpoint of the fitted left segment aligns with the starting point of the right segment. To achieve first-derivative continuity, we apply quadratic B-spline (degree of 2) on the fitted data to ensure profile regularity. As the MSE between the ODE-based and the B spline-based fittings approaches 0 (Supplementary Fig. 4F), indicating that the two lines coincide, this approach ensures continuity and is kinetically interpretable thanks to the presence of the k and b parameters. [...] To evaluate the quality of the fit, we computed the GMM threshold based on the MSE values obtained from the piecewise fitting, following the same methodology described for the monotonic fits.” Excerpt from Main Text (Discussion section):

“Although the usage of the logistic ODE has proven effective for fitting monotonic signals, our ODE-fitting approach is limited in its ability to capture peak-shaped signals ensuring regular profiles. To overcome this limitation, we developed a simplified method that merges two sigmoid functions to fit the small subset of peak-shaped signals. We further demonstrate that this approach closely approximates results obtained from B-spline fitting, while offering the added advantage of being more interpretable. A potential future direction involves fitting a sum of logistic functions which can provide a better global fit, especially in the peak region, but parameter meaning becomes more entangled. Additionally, to improve peak fitting, we suggest leveraging local minima and maxima. Finally, Gaussian distributions may be employed, as their limiting behavior is well-suited for capturing spike-like signal patterns often associated with transcriptional bursts, which are biologically relevant.”

Excerpt from new Supplementary Figure 4F:

Supplementary Fig. 4. Properties of Q1-Q4 genes. F: Distributions of MSE values computed between the ODE-based piecewise fitting and the B-spline piecewise fitting for genes exhibiting peak-like expression patterns across the three brain regions.

Ref 3.5

Reviewer's comment	l. 619-621: “We [...] further split the gene-cCRE pairs into two groups depending on whether the chromatin accessibility was positively or negatively correlated with the gene expression.” This information implies that the gene data is available. Hence, one can question the usefulness of the deep learning model that precisely aims to predict the gene profile from cCRE data.
Author's response	We appreciate the reviewer's feedback and recognize that our previous description of the enhancer and silencer models may have been unclear. To clarify, we have revised the manuscript to better explain how these models can benefit the broader genomics community. Briefly, our approach provides accurate models of both enhancer- and silencer-mediated gene regulation, enabling researchers to predict gene expression changes without requiring gene expression measurements. Generating time-series multi-omics data is costly and time-consuming,

	and only a few such datasets exist. Besides demonstrating that gene expression can be predicted over time based on chromatin accessibility, our models also offer a practical alternative to this issue: users can perform time-series experiments for a single modality (e.g., ATAC-seq or DNase-seq for chromatin accessibility) and apply our models to infer gene expression changes over time without the need for RNA-seq data. Users can apply either an enhancer or silencer model, depending on the cCREs associated with their genes of interest. There are several publicly available datasets listing enhancers and silencers in both the human and mouse genomes²⁻⁴, allowing users to cross-reference their cCREs and select the most appropriate model. Additionally, users may refine their choice based on biological considerations such as distance from the promoter, sequence motifs, and chromatin signatures characteristic of enhancers and silencers^{2,5}—all of which are independent of gene expression data. Therefore, users do not need gene expression data to apply our models. Instead, it is a missing information of interest. In summary, the partitioning of cCRE-gene pairs into positively and negatively correlated groups is done solely to optimize model accuracy, and it is not a requirement for users. By explicitly modeling both enhancers and silencers, our approach enhances gene regulation predictions over time and represents a significant advancement over state-of-the-art methods that typically focus only on enhancers (see, for instance, Mitra et al., 2024⁶).
Excerpt from manuscript	Excerpt from Main Text (Discussion section): “Our modeling framework is especially valuable given that generating time-series multi-omics data is both costly and time-consuming. Beyond demonstrating that gene expression can be predicted over time from chromatin accessibility, our approach offers a practical alternative: users can perform time-series experiments using a single modality (e.g., ATAC-seq or DNase-seq) and apply our models to infer gene expression kinetics without the need for RNA-seq data. Specifically, users can cross-reference their lists of cCREs with publicly available enhancer and silencer annotations in the human and mouse genomes^{40,54} to select the most appropriate model—enhancer or silencer—based on the regulatory elements associated with their genes of interest.”

Minor revisions:

Ref 3.6

Reviewer's comment	L.80 confusing formulation: "We numerically solve Equation 1 by optimizing the kinetic parameters k and b to reconstruct y, ": You have solved analytically the ODE to get the equation, and you need to estimate the parameter to evaluate numerically the explicit formulation of y.
Author's response	We thank the reviewer for this comment, and we agree that our previous wording was confusing. As correctly pointed out by the reviewer, our approach involves two steps. First, we solve the ODE analytically to get the explicit formulation of y . Then, we estimate the kinetic parameters k and b numerically to find the best-fitted curve to capture the original data. To make these concepts clearer, we have revised the text following the reviewer's suggestion.
Excerpt from manuscript	Excerpt from Main Text (Results section): "We first solve Equation 2 analytically to obtain an explicit formulation for y^* (see Supplementary Note , Proposition 1, for the derivation of the analytical formula) [...]. Then, we numerically estimate the kinetic parameters k^* and b^* that best fit the data."

Ref 3.7

Reviewer's comment	l. 80, please refer to supplementary note, Proposition 1, for derivation of analytical formula.
Author's response	We thank the reviewer for this comment, which we have addressed with textual changes (see excerpt below).
Excerpt from manuscript	Excerpt from Main Text (Results section): "We first solve Equation 2 analytically to obtain an explicit formulation for y^* (see Supplementary Note , Proposition 1, for the derivation of the analytical formula): $y^*(t) = \frac{b^* C^* e^{k^* t}}{b^* + C^* e^{k^* t}}, \quad \text{where} \tag{3}$ $C^* = \frac{y_{start}^*}{(1 - \frac{y_{start}^*}{b^*}) e^{k^* t_{start}}}$

Ref 3.8

Reviewer's comment	The authors need to clarify that, with their formulation, $y=0$ is an asymptote of the curve, meaning that it is never reached, except when y starts at 0. Figure 1 and several sentences (e.g. figure 1 legend) are misleading and should be clarified. Similarly, it should be stated that y_0 needs to be different from 0.
Author's response	We appreciate the reviewer's comment and agree that Figure 1 required further clarification. In the revised manuscript, we explicitly state that Figure 1 depicts the simplified form of the logistic curve, which is constrained between a lower asymptote of 0 and an upper asymptote defined by the parameter b^*. This form is a simplified case of the generalized logistic ODE (see Proposition 1 and Supplementary Fig. 2). Additionally, we have shifted t_{start} to the right in the figure to better illustrate that the starting value of the signal, y^*_{start} (previously denoted y_0), is greater than 0. As clarified in the updated figure caption, $y^*=0$ is an asymptote and is therefore never reached—i.e., y^*_{start} can only approach 0, but never equal it.
Excerpt from manuscript	Excerpt from updated Figure 1:  Figure 1: Diagram illustrating how the logistic curve can be used to model time-series genomic signals. Left panel: The curve represents a genomic signal (y axis) undergoing a logistic increase over time (x axis), modeled using the simplified form of the logistic ODE (Equation 2) and with a positive growth rate constant (i.e., $k^* > 0$). Note that the genomic signal y^* represents a translated and normalized version of the original genomic signal z. The curve is constrained between a lower asymptote of 0 and an upper asymptote defined by the parameter b^* (dashed blue line). t_{start} and t_{end} represent the initial and final time points, respectively, monitored by the experimental time course. The curve is composed of an acceleration phase followed by a deceleration phase, with the inflection point marking the transition between these phases at t_{switch} (solid red line). This curve can be used to model changes in genomic signals over time, such as gene expression (e.g., mRNA copy number measured via RNA-seq) or chromatin accessibility (e.g., number of nucleosome-free positions measured via ATAC-seq or DNase-seq). The arrows indicate how the number of mRNA copies and nucleosome-free positions increase during the acceleration and deceleration phases. Right panel: Analogous representation for a signal undergoing a logistic decrease ($k^* < 0$).

Ref 3.9

Reviewer's comment	Supp Note, Proof of proposition 1: Change the notation of c at first mention as you have not demonstrated that it is equal to the one in Proposition 1 yet. Next, you need to treat the case when $y > b$ or include the constraints $y < b$ in the proposition.
Author's response	We thank the reviewer for this comment. We have now removed the notation of c from the beginning of Proposition 1 and added the constraint $y < b$.
Excerpt from manuscript	Excerpt from Supplementary Note, Proposition 1: We show that the solution for: $\frac{dy^*}{dt} = k^* y^* \left(1 - \frac{y^*}{b^*}\right),$ where $y^* \in [0, b^*]$ (that is $b^* > y^* > 0$), and $y^* _{t=t_{start}} = y^*_{start}$, is: $y^*(t) = \frac{b^* C^* e^{k^* t}}{b^* + C^* e^{k^* t}}.$

Ref 3.10

Reviewer's comment	Equation 14: there is probably a typo as it should read the sum of squareroot of MSE, (and not square) to obtain the typical L2 norm. If this is not a typo, this would require more explanation as this would be an unusual choice of Loss.
Author's response	We thank the reviewer for their attention to this typo. The square is a typo and in the revised version is removed. In previous equation 14 (now equation 16), we just minimize the MSE (which is equivalent to minimizing the squared L2 norm of the residuals scaled by the number of samples).
Excerpt from manuscript	Excerpt from Methods section: “Next, we minimized the overall loss value by averaging across the entire batch: $Loss = \frac{1}{N} \sum_{t=1}^N (MSE_t), \tag{16}$ where N is the batch size.”

References

- [1] Motulsky, H., & Christopoulos, A. (2004). Fitting models to biological data using linear and nonlinear regression: a practical guide to curve fitting. Oxford University Press.
- [2] Doni Jayavelu, N., Jajodia, A., Mishra, A., & Hawkins, R. D. (2020). Candidate silencer elements for the human and mouse genomes. *Nat Commun*, *11*(1), 1061. <https://doi.org/10.1038/s41467-020-14853-5>
- [3] Moore, J. E., *et al.* (2024). An Expanded Registry of Candidate cis-Regulatory Elements for Studying Transcriptional Regulation. *bioRxiv*, 2024.12.26.629296. <https://doi.org/10.1101/2024.12.26.629296>
- [4] The ENCODE Project Consortium (2020). Expanded encyclopaedias of DNA elements in the human and mouse genomes. *Nature*, *583*(7818), 699–710. <https://doi.org/10.1038/s41586-020-2493-4>
- [5] Kim, S., & Wysocka, J. (2023). Deciphering the multi-scale, quantitative cis-regulatory code. *Mol Cell*, *83*(3), 373–392. <https://doi.org/10.1016/j.molcel.2022.12.032>
- [6] Mitra, S., *et al.* (2024). Single-cell multi-ome regression models identify functional and disease-associated enhancers and enable chromatin potential analysis. *Nat Genet*, *56*(4), 627–636. <https://doi.org/10.1038/s41588-024-01689-8>